# See the Emotion: A Facial Emoji Proxy Modeling for EEG Emotion Recognition

**Jingjing Hu** [1]  **Dan Guo**[*] [1 2 3]  **Haofan Cheng** [1]  **Ying Zeng** [4]  **Zhan Si** [5]  **Jinxing Zhou** [6]  **Meng Wang** [1 2 3]

## Abstract

Despite the high accuracy of EEG-based emotion recognition, existing models remain opaque "black boxes", lacking semantic grounding between abstract neural features and human-interpretable states. In this paper, we reframe EEG explainability as a cross-modal generation task, shifting the paradigm from feature attribution to behavioral visualization. We introduce Facial Emoji Proxy Modeling, a novel framework that translates high-dimensional EEG signals into identity-anonymized facial emojis. Guided by the neuroscientific inspiration of neural-facial association, this approach grounds neural representations in the manifold of observable facial dynamics. Technically, our framework integrates FMENet, a specialized backbone modeling expression-relevant spatial synergies, and the Facial Emoji Learning Branch (FELB), which treats emoji reconstruction as a structured semantic regularizer. Extensive experiments on EAV and MMER benchmarks demonstrate that our method achieves state-of-the-art accuracy among EEG-only models. Crucially, it generates semantically faithful facial animations that provide a transparent, privacy-preserving window into the brain's emotional evolution, effectively allowing users to "see the emotion" directly from neural signals. Code is available at https://github.com/xian-sh/SeeEmotion

## 1. Introduction

Decoding emotional states from electroencephalography (EEG) signals is a foundational challenge in affective computing and human-centered AI (Pillalamarri & Shanmugam, 2025). While deep learning has pushed classification accuracy to impressive levels(Zhong et al., 2022; Altaheri et al., 2023; Lawhern et al., 2018; Ding et al., 2022; Eldele et al., 2024), a profound dichotomy persists between model performance and human interpretability. In essence, prevailing models behave as inscrutable "black boxes", mapping noisy, high-dimensional EEG sequences to discrete labels without offering an *intelligible trace* of *how* the emotional experience unfolds in the neural signal. This opacity fundamentally limits their deployment in high-stakes domains, such as clinical neurofeedback and trustworthy BCI, where verifiable understanding is as critical as prediction.

The quest for interpretability has traditionally followed two paths. Post-hoc attribution methods (*e.g.*, Grad-CAM) analyze trained models to highlight influential EEG channels or time points (Miao et al., 2025; Bouazizi & Ltifi, 2025; Vakala Rani et al., 2025). However, these saliency maps remain abstract, pointing to *where* the model attends rather than explaining *what* it perceives. Alternatively, built-in neuroscientific constraints incorporate priors like spectral dynamics into network design (Zhang et al., 2021; Ning et al., 2023; Liu et al., 2024). Yet, their explanations remain couched in domain-specific jargon (*e.g.*, "alpha-band power in frontal electrodes"), failing to bridge the "last mile" to *externally observable behavioral semantics*.

The core bottleneck lies in the absence of a *semantic bridge* capable of translating internal neural dynamics into a intuitively understandable vocabulary. To address this, we propose a paradigm shift: moving from the question of "which features matter?" to "**what does this neural state look like?**". This shift is grounded in long-standing psychological and neuroscientific evidence that affective states are systematically expressed in coordinated patterns of facial muscle activity and brain dynamics. Classic emotion theories (*e.g.*, Ekman's basic emotion theory and appraisal-based models (Ekman, 1992; Scherer, 2005)) posit that internal emotional episodes are reliably coupled with prototypical facial configurations, and recent EEG-face studies further show that facial behavior is partially decodable from electrophysiological signals (Soleymani et al., 2015; Chakladar & Pal, 2024; Du et al., 2023). Our own quantitative analysis in Fig. 4 (§ 4.7) corroborates this neural-facial coupling.

[1]Hefei University of Technology [2]Institute of Artificial Intelligence, Hefei Comprehensive National Science Center [3]The Key Laboratory of Knowledge Engineering with Big Data, Hefei University of Technology [4]PLA Information Engineering University [5]University of Science and Technology of China [6]MBZUAI. Correspondence to: Dan Guo <guodan@hfut.edu.cn>.

*Proceedings of the 43rd International Conference on Machine Learning*, Seoul, South Korea. PMLR 306, 2026. Copyright 2026 by the author(s).

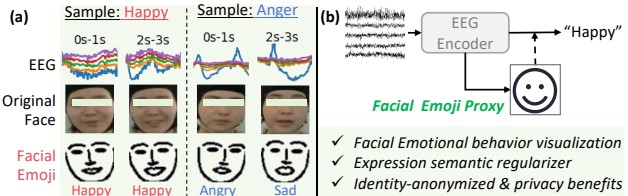

*Figure 1.* Facial Emoji Proxy Modeling. (a) Task goal: translate incomprehensible EEG signals into facial emojis that provide more clearly emotion-related cues. (b) Proposed pipeline: an EEG-based facial emoji proxy model that jointly predicts emotion labels and generates identity-anonymized emojis. Data scenarios see Fig. 8.

Motivated by above evidence, we reframe EEG-based emotion explanation as a cross-modal generation task that explicitly leverages this link (Fig. 1 a). Our model learns to translate raw EEG signals into dynamically evolving, identity-anonymized "Facial Emojis". We select emojis as visual proxies because they distill the geometric essence of expression while naturally masking away identity information, facilitating privacy (§3.3.1). This marks an interpretation object shift: from explaining feature saliency to visualizing *behavioral evolution*. The generated emoji sequence provides a direct and intuitive narrative of the brain's emotional trajectory, mitigating the "black box" dilemma through synthesis.

To realize this, we design a novel EEG–face emotion backbone, **FMENet**, explicitly structured to extract representations semantically aligned with facial dynamics. As detailed in §3.2, integrates an Expression-Relevant Spatial Merger, which aggregates EEG channels based on their functional correlation to facial movements, and a Multi-Scale Temporal Capturer that models the oscillatory patterns (Davidson et al., 2003; Liu et al., 2024) of brain dynamics. Building on this, we introduce the Facial-Emoji Learning Branch (FELB). The pipeline operates in a multi-task paradigm (Fig. 1 b): it jointly optimizes for emotion classification and identity-anonymized emoji reconstruction. By enforcing the network to reconstruct facial dynamics from EEG, FELB acts as a supervisor that aligns the neural latent space with the behavioral space. This allows our system to offer dynamic, visual explanations solely from EEG input during inference, without sacrificing and indeed enhancing accuracy.

The main contributions of this work are: (1) A Novel Paradigm for EEG Interpretability: We propose Facial Emoji Proxy Modeling, which reframes explanation as a cross-modal generation task, achieving a fundamental interpretation object shift from feature saliency to behavioral visualization. (2) Neural-Behavioral Translation Architecture: We design FMENet and FELB, a unified framework that explicitly captures expression-relevant neural dynamics and leverages facial reconstruction as a privacy-preserving semantic regularizer. (3) Empirical Validation: Extensive

experiments on EAV and MMER benchmarks demonstrate that our method achieves state-of-the-art accuracy among EEG-only models, while generating semantically faithful animations that make neural emotional dynamics directly visible and understandable.

## 2. Related Work

### 2.1. EEG-based Emotion Recognition

EEG-based emotion recognition has mainly focused on maximizing classification accuracy with increasingly complex neural architectures. Representative EEG-only models include compact convolutional networks such as EEG-Net (Lawhern et al., 2018) and TSception (Ding et al., 2022), and more recent graph- and attention-based models that explicitly model spatio-temporal structure (Li et al., 2019; Altaheri et al., 2023; Song et al., 2022; Eldele et al., 2024; Feng et al., 2022). While these approaches steadily improve performance on standard benchmarks, they are almost exclusively trained as single-task classifiers: they map high-dimensional EEG segments directly to labels, without providing an *intelligible account* of how affective dynamics unfold over time. In particular, they do not attempt to render the *behavioral correlates* of the decoded emotion (*e.g.*, corresponding facial movements), leaving a gap between neural representations and observable expression. Our framework instead couples EEG encoding with an explicit behavioral visualization objective, while maintaining competitive recognition accuracy.

### 2.2. Cross-Modal Generation for EEG Behavior Interpretation

Cross-modal generation has been used for behavior interpretability in several neighboring domains. Image-to-report models generate textual descriptions of medical images to justify predictions (Jing et al., 2018; Mohsan et al., 2022), and neural decoding studies reconstruct visual stimuli from fMRI or ECoG recordings (Nishimoto et al., 2011; Naselaris et al., 2011; St-Yves & Naselaris, 2018). Recent work even approximates static images and videos from brain activity, effectively "seeing" what a subject perceives (Ren et al., 2021; Scotti et al., 2023; Li et al., 2025). However, these efforts focus on perception or language, and rarely address *affective* dynamics from EEG. Moreover, most reconstructions are identity-preserving, which is undesirable for emotion-centric applications where identity is irrelevant and privacy is critical. Our work differs in both respects: we target affective EEG, and we introduce an emoji-based proxy that abstracts away identity while preserving the semantic geometry of facial expression, enabling privacy-preserving yet behaviorally meaningful visualization.

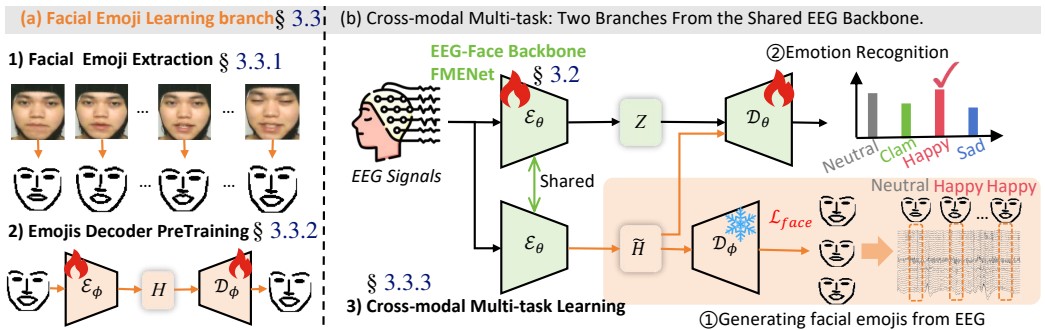

*Figure 2.* Overview of the Facial Emoji Proxy Modeling framework. (a) Facial Emoji Learning Branch: 1) facial videos are abstracted into binary, identity-anonymized Emojis and 2) learn a VAE-based emoji manifold ($\mathcal{E}_\phi$–$\mathcal{D}_\phi$), 3) training coupled with emotion recognition task. (b) Cross-modal multi-task learning: the shared FMENet backbone ($\mathcal{E}_\theta$) encodes EEG into $\mathbf{Z}$, driving two branches: ① Generative Branch maps to the emoji latent space and reconstructs dynamic emoji sequences via the frozen decoder $\mathcal{D}_\phi$, and ② Discriminative Branch predicts emotion labels conditioned on the behavior-aligned code, grounding decisions in the learned behavioral manifold.

## 3. Method

### 3.1. Problem Definition

**EEG-based Emotion Recognition.** Let $\mathbf{X} \in \mathbb{R}^{C \times T}$ denote the input EEG signals consisting of $C$ channels and $T$ time points, and let $\mathbf{Y}$ represent the corresponding emotion labels. The canonical goal is to learn a mapping function $\mathbf{X} \xrightarrow{\mathcal{E}_\theta, \mathcal{D}_\theta} \mathbf{Y}$, typically parameterized by an EEG encoder $\mathcal{E}_\theta$ and a classification head $\mathcal{D}_\theta$. Conventional approaches treat this as a purely discriminative task, optimizing parameters solely to minimize classification error. However, this "black-box" paradigm provides limited insight into the semantic structure of the learned representations.

**Cross-modal Multi-task Learning.** To transition from predictive accuracy to behavioral interpretability, we reframe the problem as a cross-modal generation task grounded in neural–facial consistency. Formally, let $\mathbf{F} \in \mathbb{R}^{M \times H \times W}$ denote a sequence of synchronized facial frames. As illustrated in Fig. 2(b), our framework establishes a dual-objective mapping: (1) Generative: for behavioral reconstruction, (2) Discriminative: for emotion recognition. Here, the generative branch $\mathbf{X} \xrightarrow{\mathcal{E}_\theta, \mathcal{D}_\phi} \mathbf{F}$ acts as a semantic regularizer, forcing the shared encoder $\mathcal{E}_\theta$ to capture latent dynamics that are not only discriminative for labels but also structurally aligned with the manifold of observable facial behavior.

### 3.2. EEG–Face Emotion Backbone

To enable robust cross-modal translation, the EEG encoder must simultaneously capture expression-relevant spatial synergies and multi-scale temporal emotion dynamics. To this end, we propose the Facial Expression-Relevant Multi-Scale Emotion Network (FMENet) (Fig. 3), which comprises two specialized modules:

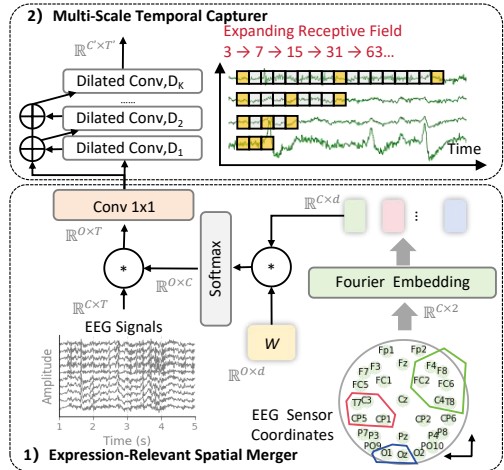

*Figure 3.* FMENet backbone, It encodes neural dynamics via two specialized modules: (1) Expression-Relevant Spatial Merger (ESM): discrete electrode coordinates are embedded by Fourier position embeddings to enable geometry-aware attention that aggregates channels into functionally coherent groups. (2) Multi-Scale Temporal Capturer (MTC): a stack of dilated 1D convolutions with exponentially expanding receptive fields captures spectro-temporal diversity, modeling both transient high-frequency responses and sustained low-frequency states.

#### 3.2.1. EXPRESSION-RELEVANT SPATIAL MERGER

EEG channels are spatially distributed over the scalp, exhibiting complex topological dependencies. To exploit neural–facial consistency at the spatial level, ESM aggregates channels based on their functional relevance to facial expressions. We encode 2D sensor locations $\mathbf{R} = \{(\mathbf{x}_q, \mathbf{y}_q)\}_{q=1}^C$ by a learnable Fourier position embedding to capture the geometric manifold of the sensor array:

$$\mathcal{F}_{\text{fourier}}(\mathbf{x}_q, \mathbf{y}_q) = \sum_{h=1}^{H} \sum_{b=1}^{H} \Big[ \mathcal{R}(\mathbf{z}_q^{(h,b)}) \cos(2\pi(h\mathbf{x}_q + b\mathbf{y}_q)) \\ + \mathcal{I}(\mathbf{z}_q^{(h,b)}) \sin(2\pi(h\mathbf{x}_q + b\mathbf{y}_q)) \Big],$$

(1)

yielding Fourier-position embeddings $\mathbf{E} \in \mathbb{R}^{C \times d}$ with learnable complex parameters $\mathbf{z}_q^{(h,b)}$, $\mathcal{R}(\cdot)$ and $\mathcal{I}(\cdot)$ represent the real and imaginary part parameters, respectively. On top of $\mathbf{E}$, we compute a soft attention mechanism to merge the discrete channels into functionally coherent spatial groups:

$$\mathbf{X}'_{o,t} = \sum_{c=1}^{C} \frac{\exp\left(\sum_{k=1}^{d} \mathbf{E}_{c,k} \mathbf{W}_{o,k}\right)}{\sum_{o'=1}^{O} \exp\left(\sum_{k=1}^{d} \mathbf{E}_{c,k} \mathbf{W}_{o',k}\right)} \cdot \mathbf{X}_{c,t}, \quad (2)$$

where $\mathbf{W} \in \mathbb{R}^{O \times d}$ is learnable and $O$ controls the number of merged channels. A subsequent $1 \times 1$ convolution produces $\mathbf{X}'' \in \mathbb{R}^{O \times T}$. In summary, ESM performs a geometry-aware, expression-driven spatial aggregation, effectively compacting the high-dimensional EEG topography into a set of latent sources that facilitate downstream cross-modal alignment.

### 3.2.2. MULTI-SCALE TEMPORAL CAPTURER

Emotion dynamics in EEG manifest across multiple frequency bands (*e.g.*, $\alpha$, $\beta$, $\gamma$), each operating at different temporal scales. To capture this spectro-temporal diversity, MTC stacks blocks of dilated 1D convolutions over the spatially aggregated signal $\mathbf{X}_1 = \mathbf{X}''$. Specifically, we employ an exponentially increasing dilation rate scheme to expand the temporal receptive field without increasing computational cost:

$$\begin{aligned}
\mathbf{Z}_k^{(1)} &= \text{Conv1D}_{1 \times 3}(\mathbf{X}_k, \text{dilation} = 2^{k \bmod D^*}) + \mathbf{X}_k, \\
\mathbf{Z}_k^{(2)} &= \text{GELU}(\text{BatchNorm}(\mathbf{Z}_k^{(1)})), \\
\mathbf{Z}_k^{(3)} &= \text{Conv1D}_{1 \times 3}(\mathbf{Z}_k^{(2)}, \text{dilation} = 2^{(k+1) \bmod D^*}), \\
\mathbf{X}_{k+1} &= \text{GLU}(\text{Conv1D}_{1 \times 1}(\mathbf{Z}_k^{(3)})) + \mathbf{Z}_k^{(1)},
\end{aligned} \quad (3)$$

where $k \in \{1, \ldots, K\}$ indexes the blocks and $D^*$ is the dilation period (*i.e.*, $D^* = 5$). By alternating dilation rates (*e.g.*, $2^{k \bmod D^*}$ and $2^{(k+1) \bmod D^*}$), the network explicitly models both transient high-frequency responses and sustained low-frequency states. The final $1 \times 1$ convolution with GLU refines channel interactions, and residual connections stabilize optimization. The final output $\mathbf{Z} = \mathbf{X}_{K+1} = \mathcal{E}_\theta(\mathbf{X}) \in \mathbb{R}^{C' \times T'}$ thus encodes a rich, multi-scale representation of emotional dynamics, serving as the shared substrate for both classification and facial generation.

### 3.3. Facial Emoji Learning Branch

The Facial Emoji Learning Branch (FELB) is the core engine of our interpretability paradigm. It operationalizes the concept of "explanation by generation" through a three-stage pipeline: (1) Abstraction: Distilling facial videos into identity-anonymized emojis; (2) Prior Learning: Pretraining a robust facial decoder; and (3) Alignment: Jointly learning to map EEG signals to this behavioral manifold. **The full three-stage training procedure of FMENet with FELB is provided in App. A.**

### 3.3.1. FACIAL EMOJI EXTRACTION

Given EEG-synchronized facial videos $V = \{v_i\}_{i=1}^{M}$ with $v_i \in \mathbb{R}^{3 \times H \times W}$, we first extract facial expression keypoints using a standard detector[1] to isolate salient regions. The resulting landmarks, including eyebrows, eyes, nose, mouth, and facial contours, are then rasterized into binary "Facial Emojis", denoted as $\mathbf{F} = \{v_i^b\}_{i=1}^{M}$ where $v_i^b \in \{0,1\}^{H \times W}$. As shown in Fig. 2(a,1), the representation offers three strategic advantages: (1) Topology Compatibility: emojis live on a regular grid and can be modeled efficiently with CNN decoders, facilitating the learning of local shape patterns; (2) Geometric Robustness: rasterization reduces sensitivity to small landmark perturbations, emphasizing the semantic geometry of expressions; and (3) Privacy by Design: texture and identity cues are removed, so explanations expose the emotional trajectory while keeping the subject anonymous.

### 3.3.2. EMOJI DECODER PRETRAINING

To obtain a stable and expressive behavioral manifold, we first pretrain a VAE-based (Kingma & Welling, 2022) emoji autoencoder on all available emojis. This provides a frozen decoder that reliably maps low-dimensional codes to plausible facial expressions. As illustrated in Fig. 2(a,2), a CNN-based encoder $\mathcal{E}_\phi$ first processes an input Facial Emoji $\mathbf{F}$ through several convolutional layers to extract hierarchical features. This is followed by a CNN decoder $\mathcal{D}_\phi$ composed of deconvolutional layers that upsample the latent code $\mathbf{H}$ to reconstruct the emoji $\hat{\mathbf{F}}$:

$$\begin{aligned}
\boldsymbol{\mu}, \boldsymbol{\sigma} &= \text{Conv}_{4\times}(\mathbf{F}), \quad \boldsymbol{\mu}, \boldsymbol{\sigma} \in \mathbb{R}^L, \\
\mathbf{H} &= \boldsymbol{\mu} + \boldsymbol{\sigma} \odot \boldsymbol{\epsilon}, \quad \boldsymbol{\epsilon} \sim \mathcal{N}(0, \mathbf{I}), \\
\hat{\mathbf{F}} &= \text{Deconv}_{4\times}(\mathbf{H}) \in \mathbb{R}^{H \times W},
\end{aligned} \quad (4)$$

where $L$ is the latent dimensionality. We optimize a binary cross-entropy reconstruction loss $\mathcal{L}_{\text{BCE}}$ between $\mathbf{F}$ and $\hat{\mathbf{F}}$ plus a KL divergence term $\mathcal{L}_{\text{KL}}$, yielding an identity-anonymized emoji latent space that will later serve as the behavioral target for EEG-to-emoji translation.

### 3.3.3. CROSS-MODAL MULTI-TASK LEARNING

In the second stage, we freeze the pretrained decoder $\mathcal{D}_\phi$ and train the EEG backbone (FMENet) under a multi-task objective, as shown in Fig. 2(b). The shared EEG representation $\mathbf{Z}$ branches into: **(1) Generative Branch:** A projection head maps $\mathbf{Z}$ into the emoji latent space: $\boldsymbol{\mu}, \boldsymbol{\sigma} = \text{Linear}_{3\times}(\text{AvgPool}(\mathbf{Z})), \boldsymbol{\mu}, \boldsymbol{\sigma} \in \mathbb{R}^L$, followed by sampling $\tilde{\mathbf{H}} = \boldsymbol{\mu} + \boldsymbol{\sigma} \odot \boldsymbol{\epsilon}$ and decoding $\tilde{\mathbf{H}}$ through the frozen $\mathcal{D}_\phi$ into an emoji sequence $\tilde{\mathbf{F}}$. This branch realizes the EEG-to-emoji behavior translation. **(2) Discriminative Branch:** The classifier $\mathcal{D}_\theta$ predicts emotion labels from the shared features, conditioned on the behavioral code, *i.e.*,

---

[1]We use dlib's `shape_predictor_68_face_landmarks`.

from $[\mathbf{Z}; \tilde{\mathbf{H}}]$. This encourages the decision boundary to align with the structure of the learned behavioral manifold.

During **training**, the total objective combines classification loss and reconstruction loss (Eq. 5). The latter reuses the VAE objective: $\mathcal{L}_{\text{face}} = \mathcal{L}_{\text{BCE}}(\tilde{\mathbf{F}}, \mathbf{F}) + \mathcal{L}_{\text{KL}}(\boldsymbol{\mu}, \boldsymbol{\sigma})$, and the classifier is optimized with a standard multi-class cross-entropy loss $\mathcal{L}_{\text{cls}}$. The overall training objective is

$$\mathcal{L}_{\text{total}} = \mathcal{L}_{\text{cls}}(\hat{\mathbf{Y}}, \mathbf{Y}) + \lambda \cdot \mathcal{L}_{\text{face}}, \tag{5}$$

where $\lambda$ balances the trade-off between discriminative power and generative fidelity. In **inference**, only EEG is required. Given $\mathbf{X}$, the model outputs both the predicted emotion label $\hat{\mathbf{Y}}$ and a sequence of reconstructed Facial Emojis. FELB thus provides an identity-anonymized, semantically faithful visualization of the brain's emotional evolution, enabling users to *see the emotion* directly from neural signals.

## 4. Experiments

### 4.1. Datasets

We evaluate our framework on two recent multimodal EEG–face datasets with synchronized recordings, **EAV** (Lee et al., 2024) and **MMER** (Yang et al., 2024), and further assess zero-shot generalization on the EEG-only **SEED** dataset (Zheng & Lu, 2015). EAV and MMER provide multi-channel EEG and frontal facial videos, enabling us to study both emotion recognition and EEG-to-emoji generation in a controlled setting, while SEED serves as a held-out target domain without facial information.

**EAV.** EAV is the first publicly available conversational EEG-audio-video dataset for emotion recognition. It contains synchronized 30-channel EEG and facial videos from 42 subjects engaged in cue-based conversations, eliciting five discrete emotions: *Neutral*, *Anger*, *Happiness*, *Sadness*, and *Calmness*. EEG recordings are sampled at 500 Hz and downsampled to 100 Hz for analysis. The dataset includes 8,400 videos (>20 s raw videos per trial) with balanced label distribution across the five classes (Fig. 11). We follow the default subject-dependent split provided by the authors, which uses a 7:3 train/test ratio across subjects. **For cross-subject experiments, we additionally adopt a fixed leave-subject-out protocol, please see App. C.3.**

**MMER.** MMER is a multimodal dataset designed for mixed-emotion recognition. It contains synchronized 18-channel EEG, galvanic skin response, photoplethysmography, and frontal facial videos recorded while participants watch emotion-eliciting video clips. EEG is recorded at 300 Hz, the dataset includes 2,336 videos (20∼30 s raw videos per trial) and provides three emotion categories at the video level: *Positive*, *Negative*, and *Mixed* (Fig. 11). Following common practice, we focus on EEG and face modalities. Since no benchmark exists for our EEG–

visual interpretation setting, we select 14 subjects (IDs {1,5,11,12,19,20,22,23,24,25,29,32,33,38}) with clearly visible facial contours, ensuring $\geq 95\%$ successful landmark detection. We adopt 20 s EEG segments aligned with the video clips, and define an 8/1/1 subject-level split into train/validation/test sets.

**SEED (zero-shot target).** SEED is a widely used EEG-based emotion recognition benchmark that contains only EEG signals, without facial videos. It consists of 62-channel EEG recorded at 200 Hz (downsampled to 100 Hz) from 15 subjects watching film clips that elicit three emotions: *Positive*, *Neutral*, and *Negative*. Following standard preprocessing, we use 4,s EEG segments. SEED is never used for training; instead, it serves purely as a held-out target dataset to evaluate zero-shot generalization of the learned EEG representations from EAV to an EEG-only setting.

### 4.2. Facial Emoji Extraction & Annotation

**Facial emoji extraction.** To obtain identity-free yet expressive visual targets, we convert each facial frame into a binary "Facial Emoji" that encodes only landmark geometry. We detect 68 facial landmarks and rasterize them onto a $56 \times 56$ grid to form $F \in \{0,1\}^{1 \times 56 \times 56}$, choosing this resolution to balance expression detail and computational cost. This yields about $4 \times 10^5$ emojis on EAV and $1.8 \times 10^4$ on MMER. Emojis are aligned to EEG at a low frame rate (one emoji per second on EAV; one emoji per 0.5 s on MMER), which is sufficient for capturing the EEG dynamics needed for emoji reconstruction while keeping sequence length manageable. The resulting emojis are used both to pretrain the VAE decoder and as supervision targets in the multi-task Generative Branch.

**Frame-level emotion annotations.** Both datasets provide only video-level emotion labels (**see Fig. 11 in App. B.1**). To better analyze the emotional content of individual facial frames and the generated emojis, we additionally derive frame-level pseudo-labels using an external facial emotion recognizer[2]. These annotations are used *solely* to evaluate whether the reconstructed emojis can sufficiently express emotions and for distributional analysis (*e.g.*, Fig. 12); they are *never* used for training or model selection.

### 4.3. Evaluation Metrics

**(1) Emotion recognition.** For all classification experiments, we report **Accuracy** and **macro F1-score**, following the EAV protocol. Macro F1 is particularly important due to label imbalance, especially on MMER and in cross-subject settings. **(2) Emoji reconstruction quality.** To assess EEG-to-emoji generation, we compute standard image-quality metrics between reconstructed emojis $\tilde{F}$ and ground-truth

---

[2]Hugging Face: dima806/facial emotions image detection

*Table 1.* **Comparison on the EAV dataset.** We report accuracy and F1-score for EEG-based emotion recognition on the EAV dataset. Methods are grouped into EEG-only baselines, multimodal (EEG+Face+Audio) models, and our proposed approach with EEG-only inference. Train and inference modalities are indicated for each method.

| Method | Ref. | Train Modality | Infer. Modality | Accuracy↑ | F1-score↑ |
|---|---|---|---|---|---|
| **EEG-Only Models** | | | | | |
| EEGNet (Lawhern et al., 2018) | JNE'18 | EEG | EEG | 60.00 | 58.00 |
| EEGformer (Wan et al., 2023) | Front. Neurosci.'23 | EEG | EEG | 53.50 | 52.00 |
| Hyper-MML(E) (Kang et al., 2025) | arXiv'25 | EEG | EEG | 71.83 | 72.04 |
| **FMENet** | Ours | EEG | EEG | 76.96 | 76.47 |
| **Multimodal Models (EEG+Face+Audio)** | | | | | |
| bc-LSTM (Ma et al., 2019) | ACM MM'19 | EEG+Face+Audio | EEG+Face+Audio | 57.21 | 57.30 |
| DialogueRNN (Majumder et al., 2019) | AAAI'19 | EEG+Face+Audio | EEG+Face+Audio | 61.20 | 61.14 |
| M3NET (Mane et al., 2020) | EMBC'20 | EEG+Face+Audio | EEG+Face+Audio | 75.14 | 75.42 |
| HAUCL (Yi et al., 2024) | ACM MM'24 | EEG+Face+Audio | EEG+Face+Audio | 75.91 | 75.92 |
| AGF-IB (Shou et al., 2024) | Inf. Fusion'24 | EEG+Face+Audio | EEG+Face+Audio | 76.19 | 76.08 |
| Hyper-MML(EFA) (Kang et al., 2025) | arXiv'25 | EEG+Face+Audio | EEG+Face+Audio | **78.21** | **77.80** |
| **Ours (EEG-Only Inference with Emoji Explanation)** | | | | | |
| **FMENet+FELB** | Ours | EEG+Face | **EEG** | 77.13 | 77.02 |

*Table 2.* **Compatibility of the FELB branch with different EEG backbones.** All backbone emotion recognition results (marked with ‡) are reproduced by open codes. **App. C.2** details each subject's performance for all backbones.

| Backbone | Ref. | Train Mod. | EAV Acc.↑ | EAV F1↑ | MMER Acc.↑ | MMER F1↑ | + FELB | Train Mod. | EAV Acc.↑ | EAV F1↑ | MMER Acc.↑ | MMER F1↑ |
|---|---|---|---|---|---|---|---|---|---|---|---|---|
| SyncNet‡ (Li et al., 2017) | NeurIPS'17 | EEG | 40.18 | 37.25 | 55.89 | 51.33 | **SyncNet+FELB** | EEG+Face | 38.92 | 36.10 | 54.46 | 50.18 |
| EEGNet‡ (Lawhern et al., 2018) | JNE'18 | EEG | 61.34 | 58.99 | 54.64 | 51.04 | **EEGNet+FELB** | EEG+Face | 59.87 | 56.35 | 52.01 | 49.92 |
| TSception‡ (Ding et al., 2022) | TAC'22 | EEG | 64.40 | 63.16 | 55.54 | 53.21 | **TSception+FELB** | EEG+Face | 62.45 | 61.94 | 54.98 | 52.05 |
| ATCNet‡ (Altaheri et al., 2023) | TII'23 | EEG | 67.68 | 66.10 | 55.89 | 51.65 | **ATCNet+FELB** | EEG+Face | 65.34 | 64.33 | 53.42 | 49.71 |
| EEGConformer‡ (Song et al., 2022) | TNSRE'23 | EEG | 63.33 | 61.94 | 66.79 | 60.08 | **EEGConformer+FELB** | EEG+Face | 62.75 | 61.20 | 65.91 | 59.34 |
| LMDA‡ (Miao et al., 2023) | NeuroImage'23 | EEG | 65.12 | 64.23 | 56.79 | 50.49 | **LMDA+FELB** | EEG+Face | 64.80 | 63.65 | 56.14 | 50.12 |
| EEGMiner‡ (Ludwig et al., 2024) | JNE'24 | EEG | 73.10 | 72.30 | 63.11 | 63.02 | **EEGMiner+FELB** | EEG+Face | 72.56 | 71.48 | 62.35 | 62.18 |
| TSLANet‡ (Eldele et al., 2024) | ICML'24 | EEG | 37.80 | 37.14 | 58.39 | 58.36 | **TSLANet+FELB** | EEG+Face | 37.25 | 36.59 | 57.62 | 57.44 |
| **FMENet** | Ours | EEG | **76.96** | **76.47** | **69.11** | **63.16** | **FMENet+FELB** | EEG+Face | **77.13** | **77.02** | **69.73** | **64.28** |

*Table 3.* **Ablation of key components in FMENet on the EAV and MMER datasets.** We report accuracy and F1-score for the full model and its variants with individual components removed.

| Method | EAV Acc.↑ | EAV F1↑ | MMER Acc.↑ | MMER F1↑ |
|---|---|---|---|---|
| **FMENet + FELB** (full model) | **77.13** | **77.02** | **69.73** | **64.28** |
| w/o FELB (§ 3.3) | 76.96 | 76.47 | 69.11 | 63.16 |
| w/o ESM module (§ 3.2.1) | 65.87 | 65.29 | 59.62 | 55.78 |
| w/o MEC module (§ 3.2.2) | 68.34 | 67.58 | 60.73 | 56.30 |

$F$: Mean Squared Error (MSE), Peak Signal-to-Noise Ratio (PSNR), and Structural Similarity (SSIM). To further quantify *semantic* fidelity, we feed the generated emojis into a ResNet-18 (He et al., 2016) classifier trained on our emoji corpus and report the resulting emoji-based emotion accuracy (Emo. Acc.) against the frame-level pseudo emotion labels derived from the facial videos (Tab. 4).

## 4.4. Implementation Details

**(1) FMENet and FELB.** Our FMENet encoder (§ 3.2) uses $H=12$ Fourier harmonics ($d=288$) for spatial encoding and merges channels into $O=8$ (EAV) or $O=6$ (MMER) latent sources. The MTC module (§ 3.2.2) stacks $K=5$ (EAV) or $K=3$ (MMER) dilated residual blocks with dilation period $D^*=5$. The facial emoji decoder $\mathcal{D}_\phi$ operates on latent codes of size $L=64$ (EAV) or $L=32$ (MMER) and outputs $56 \times 56$ emojis. The decoder is first pretrained on all emojis and then kept frozen when coupled with FMENet. **(2) Optimization.** All models are implemented in PyTorch and trained with AdamW (lr $= 5 \times 10^{-5}$), batch size 32 (EAV) or 16 (MMER), for 300 epochs on a single NVIDIA A5000 GPU. The multi-task loss uses weights $\lambda=5$ (EAV) and $\lambda=0.2$ (MMER) to balance classification and reconstruction objectives. These values are chosen in a data-driven way via a unified grid search over $[0, 10]$ (Tab. 10), which reveals broad performance plateaus ($< 3.5\%$ variation). End-to-end training per dataset finishes within 6 hours with peak memory below 16 GB. **Further hyperparameter details are in App. C.4. Model efficiency analysis is in App. C.5.**

## 4.5. Emotion Recognition Results

To evaluate our framework, we compare with state-of-the-art methods on EAV under consistent data splits. Results are in Tab. 1. **(1)** EEG-based Emotion Recognition: Our FMENet using only EEG achieves 76.96% accuracy and 76.47% F1-score, setting a new state-of-the-art. It outperforms strong baselines like Hyper-MML(E) (+5.13% Acc) and EEGNet (+16.96% Acc), showing our backbone's strength in capturing emotion-related patterns. **(2)** Distinction from Multi-modal Methods: Unlike multimodal models (*e.g.*, HAUCL, AGF-IB) that require both EEG and face inputs during in-

*Table 4.* **EEG-to-Facial Emoji reconstruction quality and semantic consistency.** All metrics are computed between the generated / baseline emojis and the *ground-truth emoji* for each trial. Metrics details see § 4.3.

| Method | EAV | | | | MMER | | | |
|---|---|---|---|---|---|---|---|---|
| | MSE↓ | PSNR↑ | SSIM↑ | Emo. Acc.↑ | MSE↓ | PSNR↑ | SSIM↑ | Emo. Acc.↑ |
| Random Emoji | 0.1543 | 8.12 dB | 0.1025 | 20.15 | 0.1628 | 7.85 dB | 0.0887 | 19.83 |
| Mean Emoji (global) | 0.0876 | 10.56 dB | 0.3241 | 35.07 | 0.0912 | 10.38 dB | 0.2983 | 34.92 |
| Class-cond. Mean Emoji | 0.0521 | 14.32 dB | 0.6124 | 61.83 | 0.0547 | 13.95 dB | 0.5879 | 60.24 |
| **Recon. Emoji (FMENet)** | **0.0191** | **21.17 dB** | **0.9138** | **80.62** | **0.0190** | **17.80 dB** | **0.8540** | **79.36** |

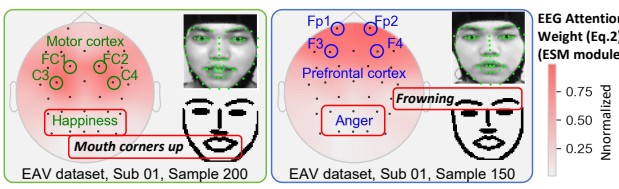

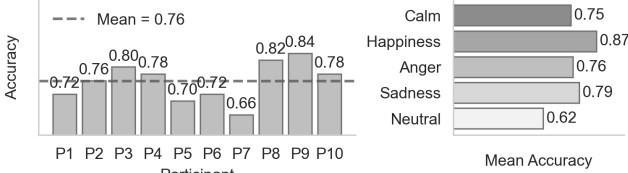

*Figure 4.* EEG spatial-attention (Eq. 2) drivers for emoji actions.

*Figure 5.* User study on EEG-reconstructed emoji comprehension.

*Table 5.* Brain region ablation via spatial attention masking on EAV dataset (5 emotions). All numbers are SSIM computed between the reconstructed and ground-truth emojis.

| Masked Region | Calm | Happiness | Anger | Sadness | Neutral | Avg. |
|---|---|---|---|---|---|---|
| **None (Full)** | **0.9105** | **0.9510** | **0.9020** | **0.8860** | **0.9195** | **0.9138** |
| Motor cortex | 0.8895 | 0.7912 | 0.8728 | 0.8791 | 0.9023 | 0.8670 |
| Prefrontal | 0.8714 | 0.9108 | 0.7426 | 0.7639 | 0.8817 | 0.8341 |

ference, our FMENet+FELB learns a direct EEG to facial emoji mapping, internalizing cross-modal association. Thus, at inference, it uses only EEG yet achieves competitive performance (77.13% Acc), even surpassing some multimodal counterparts. This shows our model successfully distills facial expression prior into EEG, bridging the cross-modal gap without simultaneous facial data. **App. C.3 further validates model's cross-subject capability.**

### 4.6. Compatibility & Key Components

**Compatibility of FELB.** We first test the compatibility of FELB by attaching the same Facial Emoji Learning Branch to a set of representative **EEG backbones (App. C.1)** from Braindecode [3] (Schirrmeister et al., 2017) on EAV and MMER (Tab. 2). For all backbones, the emoji branch weight $\lambda$ is selected from a shared grid $\lambda \in [0, 10]$ for fairness. The results show that FELB can be seamlessly integrated into diverse architectures and trained with EEG+face inputs, providing facial emoji explanations with only a modest decrease in emotion recognition accuracy. Among all models, FMENet + FELB achieves the best overall accuracy and F1 on both datasets, suggesting that our backbone is particularly well matched to the facial emoji explanation objective.

**Key Components Ablation.** We then analyze the contribution of each component in FMENet (Tab. 3). Removing ESM or MTC leads to clear drops on both EAV and MMER,

[3] https://braindecode.org

confirming their roles in capturing spatial and multi-scale emotional patterns. The variant without FELB (*w/o FELB*) further shows that the facial explanation branch brings additional gains over the EEG-only backbone, indicating that FELB provides complementary supervision for learning emotion-discriminative EEG representations.

### 4.7. Facial Emoji Reconstruction Analysis

**EEG-to-Emoji Reconstruction Quality.** Tab. 4 evaluates EEG-to-emoji reconstruction quality and semantic consistency on EAV and MMER. We compare four methods: (1) *Random Emoji*, which randomly samples an emoji from the test set; (2) *Mean Emoji (global)*, which always outputs the pixel-wise mean of all training emojis; (3) *Class-conditional Mean Emoji*, which outputs the per-class mean emoji according to the ground-truth emotion label; and (4) our EEG-driven reconstruction (FMENet). Compared with these baselines, FMENet yields substantially lower reconstruction error and higher structural similarity (*e.g.*, SSIM = 0.9138 on EAV and 0.8540 on MMER), and also achieves the highest emotion accuracy on reconstructed emojis (80.62% on EAV and 79.36% on MMER), indicating that it recovers both fine-grained facial details and emotion-consistent semantics rather than collapsing to class averages.

**Feature Analysis and Perturbation Consistency.** Fig. 4 overlays ESM (§3.2.1) spatial attention on the scalp and shows that the motor cortex is emphasized when the reconstructed emoji exhibits *mouth corner up* (*e.g.*, smiling), whereas the prefrontal cortex receives higher attention during brow-related negative expressions. To further probe whether these attentional patterns correspond to meaningful neuro–behavioural associations, we conduct a spatial perturbation analysis on EAV by masking attended brain regions during inference. Tab. 5 reports emotion-wise SSIM between reconstructed and ground-truth emojis under three

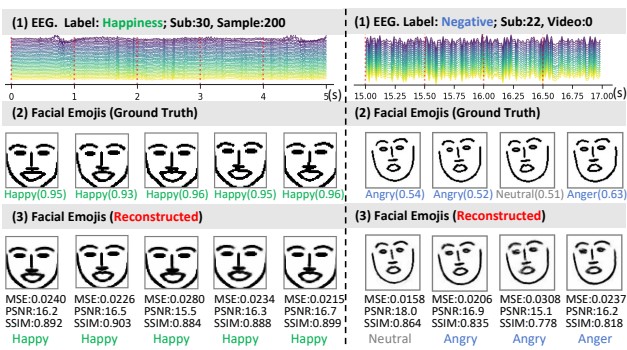

Figure 6. Emoji Visualization on EAV (left) and MMER (right).

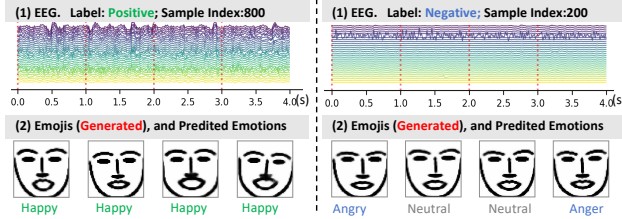

Figure 7. Model on EAV is used for SEED (No Face data).

conditions: no masking, masking motor cortex channels, and masking prefrontal channels. When masking the motor cortex, the SSIM for *Happiness* decreases from 0.9510 to 0.7912 (while remaining well above the random-emoji baseline of 0.1025 in Tab. 4); conversely, masking the prefrontal cortex has a stronger impact on *Anger* and *Sadness*. Taken together, the attention and perturbation results are consistent with prior neurophysiological findings on motor and prefrontal involvement in affective facial actions (Soleymani et al., 2015; Du et al., 2023), and suggest that FMENet has learned plausible, though indirect, associations between specific cortical regions and corresponding facial components.

**User Study on Human Comprehensibility.** We further assess whether the EEG-reconstructed emoji sequences are understandable to non-expert users. Ten participants (no neuroscience background) performed a 5-way forced-choice task on short EEG-reconstructed emoji clips reconstructed from the EAV test set: for each 5-frame sequence, they selected one of five emotions (Calm, Happiness, Anger, Sadness, Neutral). As shown in Fig. 5, the average accuracy across participants is 0.76 (chance level 0.20), with per-emotion accuracies of 0.75 (Calm), 0.87 (Happiness), 0.79 (Anger), 0.76 (Sadness), and 0.62 (Neutral). These results suggest that the reconstructed emojis provide a human-readable proxy of the underlying emotional state that can be reliably interpreted even by lay users.

**Qualitative EEG-to-Emoji Visualization.** We illustrate the interpretability of our multitask framework through qualitative EEG-to-emoji examples in Fig. 6. For both EAV and MMER, each case shows the input EEG segment, the ground-truth emoji sequence, and the corresponding reconstructions with predicted emotion labels. Across samples, the reconstructed emojis exhibit high visual fidelity (MSE $\approx 0.02$, PSNR $> 15$ dB, SSIM $\approx 0.9$), indicating that the model preserves fine-grained facial structures. On EAV, the reconstructed emojis are consistently classified as *Happy*, matching the ground-truth labels; on MMER, where annotations alternate between *Angry* and low-confidence *Neutral*, the reconstructions remain structurally stable and emotion-

ally plausible, with most predictions falling into the overall *Negative* category. **More qualitative cases are in App. E.**

### 4.8. Zero-shot Generalization to EEG-only Dataset

Since our framework requires no facial input at inference time, it can be directly applied to EEG-only datasets. We take the model trained on EAV (without any fine-tuning) and deploy it on SEED, which contains only EEG recordings. From Fig. 7, the model still produces visually and semantically plausible emoji sequences (*e.g.*, smiling faces labeled as *Happy* for positive trials, frowning faces labeled as *Angry* for negative trials). For a quantitative check, we feed the generated emojis into the ResNet-18 classifier from § 4.2 and compare the emoji-based predictions with SEED EEG labels. **As reported in App. D (Tab. 13)**, this zero-shot setup yields an average 3-way accuracy of about 60% (chance 33%), with higher scores on positive than on negative emotions. This suggests that FMENet learns a transferable EEG-to-face mapping that can drive meaningful facial animations even on datasets without any facial supervision.

## 5. Conclusion

We proposed *Facial Emoji Proxy Modeling*, a new framework that explains EEG emotion recognition by translating neural activity into dynamic, identity–agnostic facial emojis. The FMENet backbone is tailored to encode expression–relevant spatial synergies and multi–scale temporal dynamics, while the Facial Emoji Learning Branch (FELB) uses emoji reconstruction as a semantic regularizer that ties neural codes to a simple, human–readable facial manifold. Across EAV and MMER, our method achieves state–of–the–art EEG–only accuracy and generates emoji sequences that are structurally faithful, emotionally consistent, and understandable to non–experts. The same model, applied zero–shot to the EEG–only SEED dataset, still produces plausible emojis and maintains promising three–way recognition performance, indicating a transferable EEG–to–face mapping. Overall, grounding EEG representations in an emoji space offers a compact and privacy–preserving way to "see" emotional dynamics in the brain, improving both performance and interpretability for clinical and affective computing applications.

## Acknowledgement

This work is supported by Natural Science Foundation of China (72188101, 62272144), National Key R&D Program of China (NO.2024YFB3311602), the Anhui Provincial Natural Science Foundation (2408085J040), and the Major Project of Anhui Provincial Science and Technology Breakthrough Program (202423k09020001), the Anhui Provincial Graduate Quality Engineering Program (2024cxcysj002), the Fundamental Research Funds for the Central Universities (JZ2024AHST0337), the New Cornerstone Science Foundation through the XPLORER PRIZE and the CAST Young Talent Cultivation Program for Doctoral Students.

## Impact Statement

This paper introduces a novel framework for EEG-based emotion recognition using facial emoji proxies. Our primary goal is to advance interpretable affective computing for positive societal applications, such as mental health monitoring, assistive technologies for individuals with expression disorders, and empathetic human-computer interaction.

**Privacy-Preserving Interpretability.** We acknowledge that EEG signals and facial data constitute sensitive biometric information. A core motivation of our "Emoji Proxy" approach is to adhere to privacy-by-design principles: by translating neural signals into identity-anonymized emojis, our method effectively abstracts emotional expression from biometric identity. This significantly reduces the risk of re-identification compared to traditional methods that rely on raw facial video analysis.

**Dual-Use Risks and Mitigation.** While intended for assistive use, we recognize that emotion recognition technologies carry potential dual-use risks, particularly regarding non-consensual surveillance or emotional profiling in high-stakes environments (*e.g.*, employment screening). We explicitly caution against such applications without strict ethical oversight. Furthermore, our generative component produces stylized emojis rather than realistic human faces, inherently limiting the potential to generate deepfakes or misleading synthetic media.

**Ethical Compliance.** The datasets utilized in this work (EAV and MMER) were collected with informed consent of the participants and ethical approval as detailed in their respective original publications. We underscore the importance of maintaining rigorous ethical standards regarding subject consent and data privacy for future research extending this line of work.

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

This supplementary material provides additional details that could not be included in the main paper due to space constraints. We organize the content as follows:

- **(A) Training Procedure** (§ A): detailed model training pipeline and algorithmic description of FMENet with FELB.

- **(B) Datasets** (§ B): comprehensive dataset specifications, data processing, and annotation procedures.

- **(C) Extended Experiments** (§ C):
  - **(C.1) Backbone Details** (§ C.1): detailed descriptions of the selected EEG backbones used for comparison.
  - **(C.2) Performance Comparison** (§ C.2): comprehensive subject-dependent backbone comparison studies.
  - **(C.3) Cross-Subject Generalization** (§ C.3): evaluation of cross-subject generalization under leave-subject-out protocols.
  - **(C.4) Hyperparameter Sensitivity Analysis** (§ C.4): analysis of the impact of key architectural and training hyperparameters.
  - **(C.5) Model Efficiency Analysis** (§ C.5): computational efficiency analysis of different models.

- **(D) Additional Zero-Shot Analysis** (§ D): emoji-based zero-shot evaluation on the SEED dataset.

- **(E) More Case Visualizations** (§ E): more qualitative visualizations and case studies of EEG-to-emoji reconstruction.

- **(F) Impact & Ethical Considerations** (§ F): discussion of potential benefits, limitations, risks, and prohibited uses of EEG-to-emoji technologies.

- **(G) Future Work and Responsible Development** (§ G): directions for user-centered evaluation, bias auditing, richer visual proxies, and governance guidelines.

## A. Training Procedure

The complete training of FMENet with FELB follows the same three-stage pipeline as described in the main paper (§3.3): (1) identity-anonymized facial emoji construction, (2) facial emoji VAE pre-training, and (3) multi-task EEG–emoji–emotion learning. Algorithm 1 summarizes the overall procedure.

**Stage 1: Identity-Agnostic Facial Emoji Extraction.** We first convert synchronized facial videos into compact, identity-anonymized proxies. Using dlib's `shape_predictor_68_face_landmarks`, we extract 2D landmarks for each frame and rasterize them into binary facial emojis $\mathbf{F} = \{v_i^b\}_{i=1}^{M}$, where $v_i^b \in \{0, 1\}^{H \times W}$. This retains expression geometry and dynamics while discarding appearance cues, in line with our main-text definition of privacy-preserving facial emojis.

**Stage 2: Facial Emoji Decoder Pre-training.** To obtain a stable facial emoji manifold, we pre-train a VAE on all emoji frames from EAV (∼400K images). The encoder $\mathcal{E}_\phi$ maps each emoji to latent parameters $(\boldsymbol{\mu}, \boldsymbol{\sigma})$, and the decoder $\mathcal{D}_\phi$ reconstructs emojis from sampled codes. The VAE is optimized by $\mathcal{L}_{\text{face}} = \mathcal{L}_{\text{BCE}}(\hat{\mathbf{F}}, \mathbf{F}) + \mathcal{L}_{\text{KL}}(\boldsymbol{\mu}, \boldsymbol{\sigma})$, yielding a compact, expression-focused latent space that serves as the facial emoji proxy space used in FELB.

**Stage 3: Cross-modal Multi-task Learning.** In the main training stage, FMENet extracts EEG features $\mathbf{Z}_i = \mathcal{E}_\theta(\mathbf{X}_i)$, which are fed into the Facial Emoji Learning Branch (FELB) and the emotion classifier. FELB applies a lightweight three-layer linear head with temporal average pooling to predict VAE-style latent parameters $(\boldsymbol{\mu}_i, \boldsymbol{\sigma}_i)$ from $\mathbf{Z}_i$, samples latent codes $\mathbf{H}_i$, and decodes them with the frozen $\mathcal{D}_\phi$ to reconstruct emojis $\tilde{\mathbf{F}}_i$. In parallel, the emotion classification branch uses both EEG features and emoji latents, $\mathcal{D}_\theta(\mathbf{Z}_i, \mathbf{H}_i)$, to predict $\hat{\mathbf{Y}}_i$. The overall loss $\mathcal{L}_{\text{total}} = \mathcal{L}_{\text{cls}} + \lambda \cdot \mathcal{L}_{\text{face}}$ only updates $\mathcal{E}_\theta$ and $\mathcal{D}_\theta$, keeping $\mathcal{D}_\phi$ fixed. This matches the main-method design where facial emojis act as a semantic proxy: the EEG encoder is trained to produce features that are simultaneously discriminative for emotion labels and consistent with the learned facial emoji manifold.

## B. Datasets

### B.1. Data Processing and Annotation

Fig. 8 shows the data collection scenarios for this work, to enable facial emoji learning and emotion dynamics analysis, we performed comprehensive data processing and annotation on both EAV and MMER datasets. The processing pipeline ensures synchronized EEG-facial data alignment and consistent emotion labeling across modalities.

---

**Algorithm 1** Training Procedure of FMENet + FELB

---

**Require:** EEG segments $\mathbf{X} \in \mathbb{R}^{C \times T}$, emotion labels $\mathbf{Y}$, facial video frames $\mathbf{V} = \{v_i\}_{i=1}^M$, learning rate $\eta$, proxy weight $\lambda$, batch size $N_b$, maximum epochs $N_e$

**Ensure:** Trained EEG encoder $\mathcal{E}_\theta$, emotion classifier $\mathcal{D}_\theta$, facial emoji decoder $\mathcal{D}_\phi$

 1: **// Stage 1: Identity-anonymized Facial Emoji Extraction (offline)**
 2: **for** each video segment $\mathbf{V}$ **do**
 3:     Detect facial landmarks for each frame
 4:     Rasterize landmarks into binary facial emojis $\mathbf{F} = \{v_i^b\}_{i=1}^M, v_i^b \in \{0,1\}^{H \times W}$
 5: **end for**
 6: **// Stage 2: Facial emoji VAE pre-training (offline)**
 7: Train VAE encoder $\mathcal{E}_\phi$ and decoder $\mathcal{D}_\phi$ on $\mathbf{F}$
 8:     with $\mathcal{L}_{\text{face}} = \mathcal{L}_{\text{BCE}} + \mathcal{L}_{\text{KL}}$
 9: **// Stage 3: Cross-modal Multi-task Learning**
10: **for** $epoch = 1$ **to** $N_e$ **do**
11:     **for** each mini-batch $\{(\mathbf{X}_i, \mathbf{Y}_i, \mathbf{F}_i)\}_{i=1}^{N_b}$ **do**
12:         $\mathbf{Z}_i \leftarrow \mathcal{E}_\theta(\mathbf{X}_i)$ {FMENet EEG backbone}
13:         $\boldsymbol{\mu}_i, \boldsymbol{\sigma}_i \leftarrow \text{Linear}_{3\times}(\text{AvgPool}(\mathbf{Z}_i))$
14:         Sample $\boldsymbol{\epsilon} \sim \mathcal{N}(0, \mathbf{I})$, set $\mathbf{H}_i \leftarrow \boldsymbol{\mu}_i + \boldsymbol{\sigma}_i \odot \boldsymbol{\epsilon}$ {FELB latent code}
15:         $\tilde{\mathbf{F}}_i \leftarrow \mathcal{D}_\phi(\mathbf{H}_i)$ {Frozen pre-trained $\mathcal{D}_\phi$}
16:         $\mathcal{L}_{\text{face}} \leftarrow \mathcal{L}_{\text{BCE}}(\tilde{\mathbf{F}}_i, \mathbf{F}_i) + \mathcal{L}_{\text{KL}}(\boldsymbol{\mu}_i, \boldsymbol{\sigma}_i)$
17:         $\hat{\mathbf{Y}}_i \leftarrow \mathcal{D}_\theta(\mathbf{Z}_i, \mathbf{H}_i)$ {Emotion classifier with emoji-aware features}
18:         $\mathcal{L}_{\text{cls}} \leftarrow \text{CrossEntropy}(\hat{\mathbf{Y}}_i, \mathbf{Y}_i)$
19:         $\mathcal{L}_{\text{total}} \leftarrow \mathcal{L}_{\text{cls}} + \lambda \cdot \mathcal{L}_{\text{face}}$
20:         Update $\mathcal{E}_\theta, \mathcal{D}_\theta$ by gradient descent; keep $\mathcal{D}_\phi$ frozen
21:     **end for**
22: **end for**

---

### B.1.1. FACIAL EMOJI EXTRACTION

Given the synchronized facial videos provided in both datasets, we extract facial expression keypoints using dlib's `shape_predictor_68_face_landmarks` detector. The landmarks encompassing eyebrows, eyes, nose, mouth, and facial contours are rasterized into monochrome binary emojis. This representation preserves expression-related geometry while discarding identity information, serving as compact semantic proxies for emotions. Specifically, to avoid boundary effects, each emoji is extracted from the center frame of its corresponding temporal segment within the EEG window (*i.e.*, at a 0.5s offset for EAV and a 0.25s offset for MMER). The extracted emojis maintain the same file organization structure as the original facial images, with specifications detailed in Fig. 9 and Fig. 10.

### B.1.2. DATA STATISTICS AND ANNOTATION

**EAV Dataset** contains 42 subjects with balanced distribution across five emotion categories: Neutral, Anger, Happiness, Sadness, and Calmness. The original video-level emotion annotation distribution is shown in Fig. 11(a), demonstrating well-balanced class representation. After processing, we obtained approximately 400K facial frames with synchronized EEG recordings. **MMER Dataset** comprises 14 selected subjects with three emotion categories: Positive, Negative, and Mixed emotions. As illustrated in Fig. 11(b), the dataset shows a natural distribution of emotional responses to video stimuli, with mixed emotions also representing a significant portion of the data.

**Frame-level Emotion Annotation.** While both datasets provide video-level emotion labels, we further establish fine-grained frame-level annotations to capture the temporal dynamics of emotional expressions. We employ a pre-trained Vision Transformer (ViT) facial emotion recognition model from Hugging Face (`dima806/facial_emotions_image_detection`) as the facial emotion recognizer. This fine-tuned ViT-base Transformer, trained on large-scale face datasets and rigorously validated, achieves 91% overall accuracy on a 25k-image test set, with per-class F1 scores ranging from 0.85 (sad) to 0.99 (disgust), and is widely considered sufficient for serving as a semantic proxy in comparative evaluation. We use this model to annotate each facial frame with emotion probabilities. The resulting frame-level emotion distributions for both datasets are visualized in Fig. 12, revealing the subtle

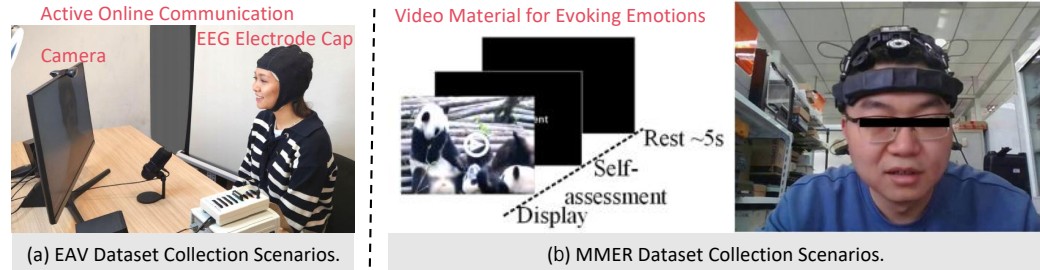

*Figure 8.* **Data collection scenarios and research goal.** (a) EAV dataset (Lee et al., 2024): active online communication with a cue-based conversational system, where 30-channel EEG and frontal facial videos are recorded while subjects engage in emotional dialogues. (b) MMER dataset (Yang et al., 2024): passive viewing of emotion-eliciting film clips, with synchronized EEG and facial videos and post-trial self-reports. We leverage these multimodal datasets to learn a dynamic mapping from incomprehensiblet EEG signals to facial expressions, enabling visualization of facial emoji proxies in EEG.

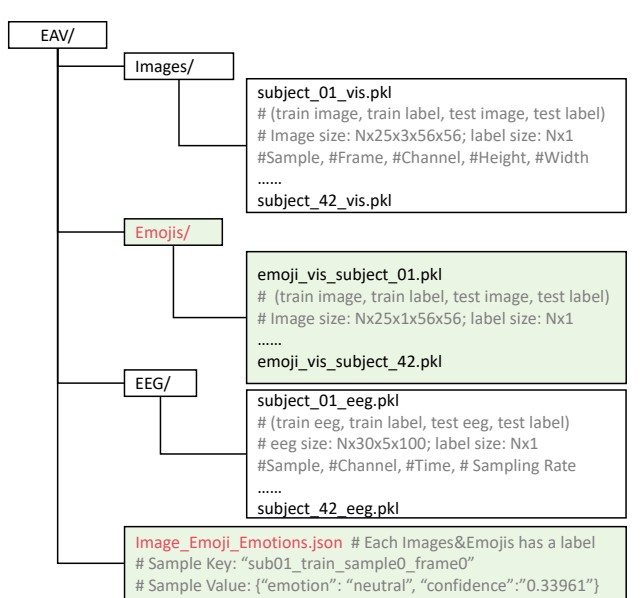

*Figure 9.* File organization of the **EAV** dataset. The data in the green area is newly extracted by us. All data will be open-sourced.

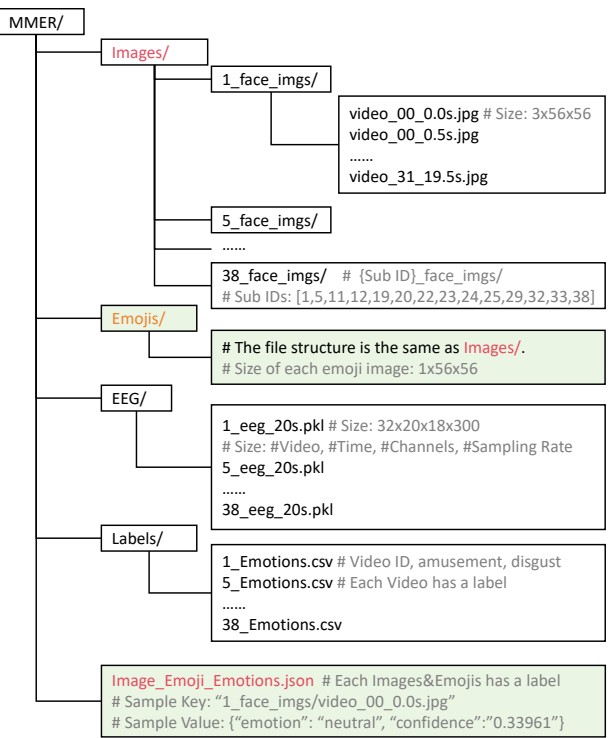

*Figure 10.* File organization of the **MMER** dataset. The data in the green area is newly extracted by us. All data will be open-sourced.

emotional transitions within each video segment. For EAV, this results in the `Image_Emoji_Emotions.json` file containing frame-level annotations with keys formatted as "`sub01_train_sample0_frame0`" and values providing emotion classification and confidence scores. Similarly for MMER, the annotation file follows the same structure with keys like "`1_face_imgs/video_00_0.0s.jpg`".

**Emoji Selection.** Each emoji is extracted from the center frame of its corresponding temporal segment within the EEG window (EAV: 0.5s offset; MMER: 0.25s offset). This avoids boundary effects. To strictly prevent any risk of cyclic verification, we emphasize that these frame-level annotations serve *only* as references for post-hoc qualitative analysis and are *never* used for model training or evaluation. Our framework's training, emoji generation, and model selection phases are entirely blind to these labels, ensuring no circular logic exists between the generated behavioral proxies and the emotional ground truth. Comparing the distributions in Fig. 11 and Fig. 12 reveals significant discrepancies between actual frame-level emotional dynamics and overall video-level emotion distributions, indicating that accurate modeling of frame-level emotional proxy emojis may face incompatibility risks with the actual video-level emotion recognition objective,

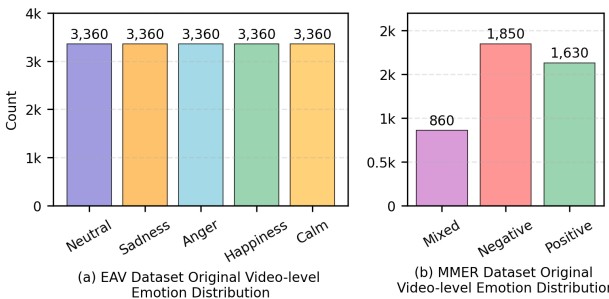

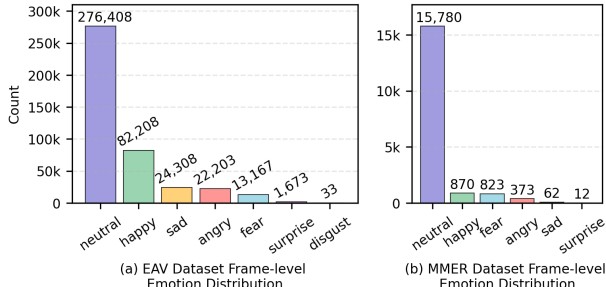

*Figure 11.* Video-level emotion annotation distribution provided by the original **EAV** and **MMER** dataset authors.

*Figure 12.* Frame-level emotion annotations created to describe the emotional dynamics of the reconstructed face emojis.

though such modeling remains necessary for constructing authentic emotional dynamics.

**Data Organization.** The processed datasets maintain consistent structures: EAV is organized by subject with separate pickle files for EEG (`subject_XX_eeg.pkl`), facial images (`subject_XX_vis.pkl`), and emojis (`emoji_vis_subject_XX.pkl`). Each file contains train/test splits with corresponding labels. MMER is structured by subject ID folders containing facial images, emojis, EEG data, and emotion labels in CSV format.

## C. Extended Experiments

In § 4.6 of the main paper, we compare our FMENet with several state-of-the-art EEG backbones from the Braindecode open-source library (Schirrmeister et al., 2017) in Tab. 2, to investigate the compatibility of different backbones with our proposed multi-task framework. This section provides (1) detailed descriptions of these backbone networks, (2) presents supplementary results for pure emotion recognition comparison (extending Tab. 2 in the main paper), (3) additional cross-subject experiments; (4) hyperparameter sensitivity analysis and (5) offers model efficiency metrics for reference.

### C.1. Backbone Details

We evaluate the following state-of-the-art EEG backbones implemented in Braindecode, all backbones are implemented using their default configurations from Braindecode and trained under identical experimental settings for fair comparison:

- **SyncNet** (Li et al., 2017) employs parameterized 1D convolutional filters inspired by Morlet wavelets, with learnable amplitude, frequency, phase, and decay parameters to extract oscillatory features from EEG signals.

- **EEGNet** (Lawhern et al., 2018) is a compact convolutional network that follows classical EEG processing pipeline: temporal frequency-selective filters, spatial filtering, and depthwise-separable convolutions, designed for efficient EEG decoding.

- **TSception** (Ding et al., 2022) combines temporal and spatial convolutional layers (Tception and Sception) with multi-scale inception windows to capture both temporal and spatial patterns in EEG data.

- **ATCNet** (Altaheri et al., 2023) utilizes a convolution-first architecture augmented with lightweight attention and temporal convolutional networks, processing overlapping temporal windows for robust EEG classification.

- **EEGConformer** (Song et al., 2022) integrates convolutional feature extraction with transformer encoders, using shallow CNN filters for initial processing followed by multi-head self-attention for long-range temporal context.

- **LMDA** (Miao et al., 2023) implements a lightweight multi-dimensional attention network with channel attention mechanisms and temporal convolution to efficiently capture EEG patterns.

- **TSLANet** (Eldele et al., 2024) is a time series lightweight adaptive network that divides input sequences into patches and employs transformer-like architecture for temporal representation learning.

*Table 6.* Supplement to Tab.2 in the main paper. Performance (%) of different backbones on **EAV** dataset.

| Subject | SyncNet‡ Acc↑ | F1↑ | EEGNet‡ Acc↑ | F1↑ | TSception‡ Acc↑ | F1↑ | ATCNet‡ Acc↑ | F1↑ | EEGConformer‡ Acc↑ | F1↑ | LMDA‡ Acc↑ | F1↑ | EEGMiner‡ Acc↑ | F1↑ | TSLANet‡ Acc↑ | F1↑ | **FMENet (Ours)** Acc↑ | F1↑ |
|---|---|---|---|---|---|---|---|---|---|---|---|---|---|---|---|---|---|---|
| 1 | 52.50 | 48.19 | 53.50 | 50.56 | 67.50 | 66.58 | 80.00 | 80.06 | 72.50 | 72.61 | 72.50 | 72.42 | 65.00 | 65.73 | 27.50 | 27.84 | 87.50 | 87.35 |
| 2 | 60.00 | 58.47 | 97.50 | 97.48 | 92.50 | 92.62 | 90.00 | 89.44 | 97.50 | 97.48 | 87.50 | 87.17 | 97.50 | 97.48 | 35.00 | 35.34 | 92.50 | 92.59 |
| 3 | 65.00 | 61.49 | 76.00 | 75.09 | 67.50 | 64.50 | 85.00 | 84.59 | 70.00 | 67.67 | 72.50 | 70.28 | 72.50 | 70.28 | 27.50 | 29.75 | 90.00 | 90.04 |
| 4 | 45.00 | 42.81 | 73.50 | 72.15 | 72.50 | 71.72 | 72.50 | 71.02 | 77.50 | 77.69 | 62.50 | 61.77 | 67.50 | 68.20 | 42.50 | 40.33 | 77.50 | 76.75 |
| 5 | 20.00 | 18.47 | 68.50 | 67.77 | 50.00 | 49.39 | 70.00 | 70.15 | 57.50 | 57.11 | 55.00 | 52.34 | 57.50 | 56.12 | 32.50 | 28.31 | 65.00 | 64.32 |
| 6 | 42.50 | 40.82 | 78.50 | 75.75 | 60.00 | 53.18 | 67.50 | 63.67 | 72.50 | 72.22 | 72.50 | 71.48 | 87.50 | 87.20 | 40.00 | 40.02 | 80.00 | 79.89 |
| 7 | 50.00 | 48.17 | 61.00 | 58.93 | 70.00 | 69.11 | 75.00 | 74.01 | 80.00 | 80.88 | 75.00 | 75.57 | 75.00 | 74.71 | 47.50 | 43.34 | 85.00 | 84.43 |
| 8 | 40.00 | 40.42 | 53.50 | 52.18 | 67.50 | 65.35 | 67.50 | 64.97 | 57.50 | 59.00 | 75.00 | 74.37 | 70.00 | 69.27 | 45.00 | 44.38 | 77.50 | 76.24 |
| 9 | 32.50 | 30.14 | 56.50 | 55.95 | 57.50 | 58.24 | 60.00 | 60.06 | 50.00 | 46.02 | 55.00 | 55.77 | 62.50 | 61.57 | 37.50 | 37.42 | 70.00 | 69.60 |
| 10 | 40.00 | 36.24 | 60.00 | 59.07 | 60.00 | 59.90 | 72.50 | 72.40 | 60.00 | 59.74 | 65.00 | 65.39 | 77.50 | 76.58 | 32.50 | 31.67 | 77.50 | 77.62 |
| 11 | 35.00 | 33.97 | 58.50 | 57.70 | 67.50 | 67.43 | 45.00 | 45.85 | 52.50 | 53.00 | 67.50 | 66.15 | 82.50 | 82.49 | 40.00 | 40.79 | 62.50 | 60.23 |
| 12 | 30.00 | 27.29 | 33.50 | 32.22 | 57.50 | 52.50 | 65.00 | 63.04 | 55.00 | 53.35 | 57.50 | 52.21 | 67.50 | 64.78 | 50.00 | 46.41 | 60.00 | 52.27 |
| 13 | 27.50 | 22.65 | 55.00 | 54.19 | 50.00 | 48.21 | 70.00 | 69.42 | 47.50 | 45.86 | 52.50 | 48.30 | 62.50 | 62.11 | 40.00 | 40.62 | 77.50 | 77.39 |
| 14 | 35.00 | 29.09 | 41.00 | 39.12 | 55.00 | 54.96 | 55.00 | 54.69 | 50.00 | 45.94 | 62.50 | 62.64 | 72.50 | 73.26 | 30.00 | 26.86 | 67.50 | 65.47 |
| 15 | 42.50 | 35.54 | 50.00 | 49.60 | 50.00 | 46.82 | 57.50 | 56.20 | 65.00 | 64.91 | 60.00 | 60.12 | 65.00 | 65.48 | 40.00 | 38.42 | 72.50 | 73.05 |
| 16 | 45.00 | 43.91 | 47.50 | 43.42 | 67.50 | 65.69 | 67.50 | 64.97 | 65.00 | 59.93 | 60.00 | 56.04 | 65.00 | 62.85 | 45.00 | 45.83 | 65.00 | 63.92 |
| 17 | 22.50 | 16.29 | 77.50 | 75.49 | 80.00 | 78.65 | 85.00 | 84.33 | 82.50 | 82.12 | 80.00 | 79.01 | 85.00 | 85.26 | 42.50 | 43.48 | 90.00 | 89.79 |
| 18 | 27.50 | 26.73 | 77.50 | 76.15 | 70.00 | 68.96 | 87.50 | 87.35 | 80.00 | 80.41 | 72.50 | 73.65 | 85.00 | 84.87 | 42.50 | 42.00 | 87.50 | 87.57 |
| 19 | 52.50 | 51.13 | 82.00 | 80.17 | 65.00 | 65.02 | 80.00 | 79.26 | 57.50 | 53.06 | 72.50 | 71.47 | 85.00 | 84.87 | 35.00 | 35.07 | 87.50 | 87.33 |
| 20 | 45.00 | 41.15 | 67.50 | 67.50 | 75.00 | 73.77 | 75.00 | 73.67 | 70.00 | 67.12 | 80.00 | 79.27 | 87.50 | 87.55 | 32.50 | 33.17 | 87.50 | 87.41 |
| 21 | 47.50 | 46.95 | 73.50 | 72.24 | 75.00 | 75.37 | 72.50 | 68.98 | 65.00 | 63.50 | 70.00 | 70.16 | 85.00 | 85.08 | 35.00 | 34.49 | 82.50 | 81.98 |
| 22 | 25.00 | 11.22 | 70.00 | 69.65 | 67.50 | 67.86 | 67.50 | 65.13 | 72.50 | 72.62 | 60.00 | 58.13 | 60.00 | 53.39 | 45.00 | 46.97 | 70.00 | 68.40 |
| 23 | 37.50 | 38.49 | 55.40 | 53.32 | 62.50 | 57.61 | 62.50 | 60.99 | 40.00 | 36.52 | 45.00 | 46.85 | 60.00 | 57.91 | 35.00 | 35.63 | 75.00 | 75.48 |
| 24 | 57.50 | 55.46 | 35.00 | 31.11 | 87.50 | 86.63 | 77.50 | 73.01 | 80.00 | 78.52 | 67.50 | 65.88 | 77.50 | 77.56 | 37.50 | 36.02 | 85.00 | 85.11 |
| 25 | 35.00 | 23.08 | 61.00 | 59.01 | 55.00 | 55.25 | 65.00 | 61.51 | 67.50 | 63.07 | 47.50 | 47.51 | 62.50 | 60.51 | 37.50 | 36.23 | 65.00 | 65.75 |
| 26 | 30.00 | 28.56 | 53.00 | 48.56 | 67.50 | 65.00 | 65.00 | 63.95 | 52.50 | 52.29 | 62.50 | 61.53 | 67.50 | 65.59 | 45.00 | 43.36 | 85.00 | 84.82 |
| 27 | 35.00 | 34.13 | 60.00 | 58.71 | 82.50 | 82.57 | 55.00 | 53.24 | 42.50 | 42.57 | 82.50 | 82.48 | 85.00 | 85.20 | 37.50 | 37.48 | 80.00 | 80.32 |
| 28 | 62.50 | 62.26 | 88.50 | 87.33 | 80.00 | 79.60 | 82.50 | 81.84 | 82.50 | 82.49 | 82.50 | 82.69 | 90.00 | 89.92 | 40.00 | 40.95 | 87.50 | 87.55 |
| 29 | 40.00 | 32.92 | 36.20 | 31.25 | 37.50 | 38.14 | 42.50 | 37.68 | 40.00 | 32.72 | 32.50 | 30.91 | 45.00 | 43.09 | 30.00 | 28.48 | 52.50 | 50.13 |
| 30 | 47.50 | 44.59 | 51.00 | 55.13 | 60.00 | 59.53 | 57.50 | 56.04 | 55.00 | 53.03 | 67.50 | 67.88 | 60.00 | 59.57 | 40.00 | 38.65 | 62.50 | 62.64 |
| 31 | 35.00 | 36.10 | 52.00 | 47.65 | 62.50 | 62.76 | 60.00 | 60.06 | 52.50 | 49.77 | 52.50 | 53.17 | 72.50 | 71.61 | 32.50 | 32.52 | 75.00 | 75.38 |
| 32 | 32.50 | 31.26 | 57.00 | 54.76 | 62.50 | 62.69 | 72.50 | 70.33 | 77.50 | 77.35 | 55.00 | 54.60 | 72.50 | 71.37 | 42.50 | 42.08 | 72.50 | 71.94 |
| 33 | 30.00 | 21.23 | 87.50 | 81.59 | 77.50 | 76.98 | 80.00 | 80.52 | 85.00 | 84.56 | 87.50 | 87.49 | 77.50 | 77.72 | 35.00 | 35.17 | 92.50 | 92.22 |
| 34 | 27.50 | 27.15 | 45.00 | 43.73 | 50.00 | 48.50 | 40.00 | 34.59 | 50.00 | 49.49 | 45.00 | 44.50 | 70.00 | 68.56 | 40.00 | 36.11 | 77.50 | 78.49 |
| 35 | 40.00 | 38.92 | 43.60 | 36.77 | 42.50 | 42.74 | 37.50 | 37.81 | 32.50 | 30.62 | 52.50 | 52.36 | 57.50 | 55.77 | 37.50 | 37.68 | 65.00 | 63.95 |
| 36 | 42.50 | 40.06 | 75.00 | 73.71 | 72.50 | 70.68 | 80.00 | 79.38 | 57.50 | 52.42 | 75.00 | 74.57 | 77.50 | 76.21 | 47.50 | 44.51 | 85.00 | 84.97 |
| 37 | 40.00 | 39.10 | 68.50 | 61.87 | 57.50 | 53.80 | 65.00 | 60.68 | 62.50 | 63.21 | 52.50 | 49.32 | 60.00 | 56.71 | 35.00 | 35.92 | 65.00 | 64.98 |
| 38 | 52.50 | 50.89 | 57.00 | 51.93 | 65.00 | 61.14 | 80.00 | 77.39 | 80.00 | 76.85 | 77.50 | 75.75 | 87.50 | 86.97 | 30.00 | 29.52 | 92.50 | 92.53 |
| 39 | 57.50 | 57.03 | 58.50 | 56.72 | 60.00 | 58.59 | 72.50 | 71.26 | 65.00 | 64.53 | 62.50 | 62.32 | 67.50 | 66.46 | 32.50 | 34.22 | 77.50 | 77.77 |
| 40 | 32.50 | 27.00 | 36.00 | 34.20 | 50.00 | 47.92 | 50.00 | 46.46 | 50.00 | 48.93 | 55.00 | 54.80 | 80.00 | 80.77 | 35.00 | 34.54 | 67.50 | 66.99 |
| 41 | 27.50 | 26.54 | 42.00 | 37.45 | 50.00 | 49.53 | 45.00 | 41.51 | 42.50 | 42.55 | 70.00 | 69.78 | 72.50 | 71.99 | 37.50 | 37.34 | 70.00 | 69.78 |
| 42 | 42.50 | 38.46 | 90.00 | 90.22 | 77.50 | 77.07 | 85.00 | 84.86 | 87.50 | 87.55 | 75.00 | 69.67 | 80.00 | 79.63 | 32.50 | 30.91 | 87.50 | 87.21 |
| **Avg.(↑)** | 40.18 | 37.25 | 61.34 | 58.99 | 64.40 | 63.16 | 67.68 | 66.10 | 63.33 | 61.94 | 65.12 | 64.23 | 73.10 | 72.30 | 37.80 | 37.14 | **76.96** | **76.47** |
| **Std.(↓)** | 11.13 | 12.32 | 16.15 | 16.72 | 11.92 | 12.21 | 13.48 | 14.08 | 15.10 | 15.90 | 12.42 | 12.66 | 11.33 | 12.08 | 5.58 | 5.40 | **10.40** | **11.08** |

*Table 7.* Supplement to Tab.2 in the main paper. Performance (%) of different backbones on **MMER** dataset.

| Subject | SyncNet‡ Acc↑ | F1↑ | EEGNet‡ Acc↑ | F1↑ | TSception‡ Acc↑ | F1↑ | ATCNet‡ Acc↑ | F1↑ | EEGConformer‡ Acc↑ | F1↑ | LMDA‡ Acc↑ | F1↑ | EEGMiner‡ Acc↑ | F1↑ | TSLANet‡ Acc↑ | F1↑ | **FMENet (Ours)** Acc↑ | F1↑ |
|---|---|---|---|---|---|---|---|---|---|---|---|---|---|---|---|---|---|---|
| 1 | 57.50 | 55.23 | 60.00 | 59.04 | 57.50 | 56.62 | 67.50 | 67.33 | 72.50 | 74.76 | 52.50 | 42.62 | 89.00 | 89.97 | 62.50 | 63.89 | 90.00 | 92.06 |
| 5 | 72.50 | 63.04 | 70.00 | 61.76 | 82.50 | 78.70 | 72.50 | 63.04 | 75.00 | 64.29 | 75.00 | 64.29 | 82.50 | 83.61 | 77.50 | 69.76 | 77.50 | 69.76 |
| 11 | 35.00 | 38.49 | 35.00 | 38.57 | 42.50 | 47.36 | 32.50 | 34.29 | 72.50 | 63.04 | 52.50 | 54.31 | 62.50 | 68.77 | 60.00 | 66.09 | 70.00 | 71.43 |
| 12 | 52.50 | 56.45 | 55.00 | 61.36 | 52.50 | 57.37 | 35.00 | 41.86 | 65.00 | 61.72 | 52.50 | 55.92 | 40.00 | 42.18 | 52.50 | 57.06 | 72.50 | 63.04 |
| 19 | 60.00 | 50.81 | 57.50 | 52.06 | 52.50 | 42.56 | 57.50 | 46.43 | 60.00 | 53.21 | 57.50 | 49.81 | 35.00 | 33.96 | 52.50 | 41.69 | 50.00 | 33.33 |
| 20 | 45.00 | 47.37 | 45.00 | 49.47 | 40.00 | 49.55 | 57.50 | 61.71 | 67.50 | 64.51 | 50.00 | 53.00 | 52.50 | 58.62 | 52.50 | 61.93 | 52.50 | 55.68 |
| 22 | 57.50 | 48.13 | 52.50 | 39.24 | 60.00 | 58.14 | 52.50 | 49.89 | 55.00 | 47.59 | 50.00 | 33.33 | 60.00 | 55.08 | 52.50 | 42.62 | 50.00 | 33.33 |
| 23 | 75.00 | 75.00 | 72.50 | 72.00 | 67.50 | 66.91 | 72.50 | 68.97 | 75.00 | 64.29 | 70.00 | 61.76 | 72.50 | 68.49 | 65.00 | 65.00 | 75.00 | 64.29 |
| 24 | 60.00 | 53.03 | 62.50 | 62.84 | 57.50 | 63.01 | 60.00 | 55.19 | 65.00 | 61.54 | 55.00 | 60.95 | 47.50 | 46.71 | 65.00 | 68.94 | 62.50 | 64.58 |
| 25 | 52.50 | 46.89 | 47.50 | 47.20 | 50.00 | 48.52 | 50.00 | 50.13 | 52.50 | 38.66 | 62.50 | 56.36 | 72.50 | 74.36 | 57.50 | 57.58 | 67.50 | 66.47 |
| 29 | 55.00 | 44.18 | 55.00 | 47.59 | 57.50 | 50.55 | 52.50 | 38.66 | 80.00 | 84.01 | 62.50 | 57.04 | 81.50 | 82.91 | 62.50 | 63.64 | 100.00 | 100.00 |
| 32 | 72.50 | 63.04 | 72.50 | 63.04 | 72.50 | 63.04 | 72.50 | 63.04 | 75.00 | 64.29 | 75.00 | 64.29 | 72.50 | 72.53 | 55.00 | 57.14 | 75.00 | 64.29 |
| 33 | 37.50 | 34.12 | 27.50 | 18.92 | 35.00 | 29.23 | 47.50 | 44.11 | 65.00 | 56.44 | 27.50 | 14.75 | 56.50 | 47.96 | 47.50 | 47.37 | 72.50 | 67.56 |
| 38 | 50.00 | 42.87 | 52.50 | 41.44 | 50.00 | 33.33 | 52.50 | 38.44 | 55.00 | 42.82 | 52.50 | 38.44 | 59.00 | 57.18 | 55.00 | 54.39 | 52.50 | 38.44 |
| **Avg.(↑)** | 55.89 | 51.33 | 54.64 | 51.04 | 55.54 | 53.21 | 55.89 | 51.65 | 66.79 | 60.08 | 56.79 | 50.49 | 63.11 | 63.02 | 58.39 | 58.36 | **69.11** | **63.16** |
| **Std.(↓)** | 12.11 | 10.77 | 13.15 | 13.75 | 12.72 | 13.14 | 12.73 | 11.60 | 8.68 | 11.92 | 12.19 | 13.94 | 16.17 | 16.91 | 7.70 | 9.13 | **14.89** | **19.19** |

## C.2. Performance Comparison

Tabs. 6&7 provide detailed subject-level performance comparison on the EAV and MMER datasets, extending the aggregated results in Tab.2 of the main paper. The comprehensive per-subject analysis reveals several important observations:

- **Individual Variability:** Significant performance variations exist across different subjects, highlighting the challenge of inter-subject differences in EEG-based emotion recognition. For instance, SyncNet shows particularly low performance on subjects 5 (20.00% Acc) and 17 (22.50% Acc), while achieving relatively better results on subject 2 (60.00% Acc).

- **Consistent Superiority:** Our FMENet demonstrates robust performance across most subjects, achieving the highest or competitive accuracy in 38 out of 42 subjects. Notably, it maintains strong performance on challenging subjects where other methods struggle, such as subject 15 (72.50% vs. SyncNet's 42.50%) and subject 13 (77.50% vs. SyncNet's 27.50%).

- **Method-specific Patterns:** Different backbones exhibit distinct performance characteristics. EEGMiner shows strong overall performance but suffers from instability on certain subjects (*e.g.*, 45.00% on subject 29), while TSLANet demonstrates the most inconsistent performance across subjects, ranging from 27.50% to 50.00% accuracy.

**Further explanation for Tab.2 of main paper.** The consistent degradation observed when integrating FELB with existing backbones (Tab.2 of main paper), combined with the detailed per-subject analysis in Tabs. 6&7, highlights the unique compatibility of our FMENet architecture with the facial explanation task. While most backbones experience performance drops when adding the facial reconstruction objective, FMENet not only maintains but slightly improves accuracy (76.96% $\rightarrow$ 77.13% on EAV), demonstrating its ability to effectively balance the dual demands of classification accuracy and interpretable representation learning. **The performance degradation is understandable for other backbones** given the significant discrepancies between frame-level emotional dynamics and video-level emotion distributions revealed in Fig. 11 and Fig. 12. The inherent **incompatibility risks** between accurate frame-level emotional proxy emoji modeling and video-level emotion recognition objectives make this multi-task optimization particularly challenging. Most existing backbones, designed primarily for single-task classification, struggle to reconcile these conflicting objectives.

However, our FMENet is **specifically designed to address this challenge** through its Expression-Relevant Spatial Merger and Multi-Scale Temporal Capturer modules, which learn representations that simultaneously capture emotion-related patterns for classification and expression-related dynamics for face reconstruction. This architectural design enables FMENet to navigate the complex trade-offs between these objectives, achieving interpretability without sacrificing recognition performance.

## C.3. Cross-Subject Generalization

For the cross-subject evaluation in Tab. 8, we adopt a leave-subject-out protocol with a fixed random seed (42). The held-out test subjects are: EAV, [6, 7, 9, 15, 16, 18, 28, 41]; MMER, [5, 29]. All competing methods are trained and evaluated under exactly the same splits to ensure a fair comparison.

Tab. 8 summarizes the cross-subject EEG-based emotion recognition results on EAV and MMER. Across all baselines, FMENet achieves the best or second-best performance on both datasets, indicating strong invariance to subject-specific variations when using EEG alone. Notably, the FMENet+FELB variant, which introduces identity-anonymized Emoji Face priors only during training, consistently matches or slightly surpasses the EEG-only FMENet in terms of both accuracy and F1-score. This demonstrates that the proposed facial prior acts as a robust, complementary regularizer that enhances cross-subject generalization while preserving the purely EEG-based inference setting.

## C.4. Hyperparameter Sensitivity Analysis

To complement the implementation details in § 3.2 and § 3.2.2, we further analyze the sensitivity of FMENet to several key architectural hyperparameters, as summarized in Tab. 9. For each dataset, we vary the number of spatial output heads $O$, the convolutional depth $K$ of the MTC module, and the latent dimension $L$ of the facial emoji decoder, while keeping all other settings fixed to the default configuration described in the main text.

The resulting trends are consistent with the design choices adopted in Eq. 5. For **EAV**, the best performance is achieved with $O=8$ spatial mergers and $K=5$ dilated residual blocks, which provide sufficient capacity to capture richer spatio–temporal

*Table 8.* **Cross-subject EEG-based emotion recognition on the EAV and MMER datasets.** We report accuracy and F1-score for EEG-only models and our proposed approach. All methods use EEG as the inference modality. ‡ denotes reproduced results using open-source implementations.

| Method | Train Modality | EAV | | MMER | |
|---|---|---|---|---|---|
| | | Accuracy↑ | F1-score↑ | Accuracy↑ | F1-score↑ |
| **EEG-Only Models** | | | | | |
| SyncNet‡ (Li et al., 2017) | EEG | 32.94 | 29.98 | 49.35 | 50.91 |
| EEGNet‡ (Lawhern et al., 2018) | EEG | 36.39 | 34.26 | 55.81 | 54.93 |
| TSception‡ (Ding et al., 2022) | EEG | 34.67 | 33.93 | 68.32 | 58.56 |
| ATCNet‡ (Altaheri et al., 2023) | EEG | 38.19 | 34.59 | 65.32 | 58.95 |
| EEGConformer‡ (Song et al., 2022) | EEG | 35.75 | 29.12 | 64.35 | 59.74 |
| LMDA‡ (Miao et al., 2023) | EEG | 38.53 | 35.89 | 66.13 | 59.77 |
| EEGMiner‡ (Ludwig et al., 2024) | EEG | 34.47 | 33.49 | 54.03 | 56.37 |
| TSLANet‡ (Eldele et al., 2024) | EEG | 30.44 | 30.07 | 51.77 | 52.64 |
| **FMENet** | EEG | **38.86** | **37.83** | **68.87** | **60.63** |
| **Ours (EEG-Only Inference with Emoji Explanation)** | | | | | |
| **FMENet+FELB** | EEG+Face | **39.05** | **38.17** | **69.23** | **61.50** |

*Table 9.* Ablation on key architectural and sampling hyperparameters for EAV and MMER. We vary: (i) the output head dimension $O$ (Eq. 2), (ii) the convolutional depth $K$ of the MTC module (Eq. 3), (iii) the latent dimension $L$ of the facial emoji decoder (Eq. 4), and (iv) the temporal interval $\Delta t$ represented by each emoji (§ 4.2).

| Component / Hyperparameter | EAV | | MMER | |
|---|---|---|---|---|
| | Value | Accuracy (%) | Value | Accuracy (%) |
| Output head $O$ | 4 | 71.50 | 4 | 61.20 |
| | 6 | 75.50 | 6 | **69.73** |
| | 8 | **77.13** | 8 | 66.20 |
| | 16 | 73.20 | 10 | 65.80 |
| | 20 | 74.10 | 16 | 61.50 |
| Conv depth $K$ | 1 | 70.50 | 1 | 67.50 |
| | 3 | 75.10 | 3 | **69.73** |
| | 5 | **77.13** | 5 | 61.20 |
| | 7 | 73.80 | 7 | 65.10 |
| Latent dim $L$ | 16 | 73.80 | 16 | 63.20 |
| | 32 | 72.50 | 32 | **69.73** |
| | 64 | **77.13** | 64 | 62.50 |
| | 128 | 72.90 | 128 | 64.90 |
| | 256 | 74.60 | 256 | 64.37 |
| Temporal interval $\Delta t$ (s) per emoji | 0.2 | 73.01 | 0.2 | 69.01 |
| | 0.5 | 76.00 | 0.5 | **69.73** |
| | 1.0 | **77.13** | 1.0 | 67.41 |
| | 1.5 | 72.25 | 1.5 | 65.80 |
| | 2.0 | 73.41 | 2.0 | 64.62 |

*Table 10.* Loss-weight ablation for the multi-task coefficient $\lambda$ (Eq. 5). Accuracies are reported for a unified grid search on both datasets, corresponding to the settings description in § 4.4.

| Dataset | Loss weight $\lambda$ (grid-search point) | | | | | | | | | |
|---|---|---|---|---|---|---|---|---|---|---|
| | 0 | 0.1 | 0.2 | 0.5 | 1 | 2 | 3 | 5 | 7 | 10 |
| EAV Acc (%) | 73.8 | 74.7 | 73.76 | 76.26 | 74.2 | 75.76 | 76.38 | **77.1** | 76.01 | 75.4 |
| $\Delta$% vs best | -3.3 | -2.4 | -3.34 | -0.84 | -2.9 | -1.34 | -0.72 | best | -1.09 | -1.7 |
| MMER Acc (%) | 68.9 | 66.8 | **69.7** | 66.8 | 66.3 | 68.66 | 67.94 | 68.84 | 69.02 | 66.87 |
| $\Delta$% vs best | -0.8 | -2.9 | best | -2.9 | -3.4 | -1.04 | -1.76 | -0.86 | -0.68 | -2.83 |

dynamics in actively expressed, strong emotions. For **MMER**, a more compact configuration with $O{=}6$ and $K{=}3$ already achieves the highest accuracy, indicating that its passively elicited, weaker emotional responses can be modeled with shallower temporal processing. The latent size $L$ in the facial emoji reconstruction branch is set to 64 for EAV and 32 for MMER, striking a balance between representational capacity and computational efficiency across datasets.

We also investigate the temporal aggregation interval $\Delta t$ encoded by each facial emoji token. Tab. 9 shows that, for **EAV**, the highest accuracy is obtained when each emoji summarizes approximately $\Delta t{=}1.0$ s of EEG (77.13%), while shorter

(0.2–0.5 s) or longer (1.5–2.0 s) intervals lead to slightly lower but comparable performance. For **MMER**, the best result is observed at $\Delta t$=0.5 s (69.73%), with a gradual decrease when moving to either finer (0.2 s) or coarser (1.0–2.0 s) temporal resolutions. Overall, the variation across different $\Delta t$ choices remains moderate on both datasets, indicating that FMENet is reasonably stable with respect to the temporal granularity used for constructing emoji tokens, with dataset-specific optima around 1.0 s for EAV and 0.5 s for MMER.

We additionally study the impact of the multi–task loss weight $\lambda$, which balances the classification objective and the facial emoji reconstruction objective, in Tab. 10. The search is performed over a unified grid $\lambda \in [0, 10]$ for both datasets. Consistent with the default settings in § 4.4, the best-performing weights are $\lambda$=5 for EAV and $\lambda$=0.2 for MMER, respectively. Importantly, the performance curves exhibit broad plateaus (within 3.5% relative variation) rather than sharp peaks, suggesting that FMENet is reasonably robust to moderate changes of $\lambda$. Strong-emotion EAV (active expression) benefits from heavier facial regularization, whereas weak-emotion MMER (passive elicitation) prefers lighter coupling, which is in line with reported neuro–facial alignment patterns (Du et al., 2023; Soleymani et al., 2015; Ding et al., 2022) and further supports the design choices reported in § 4.4.

## C.5. Model Efficiency Analysis

We further assess the practical deployment potential through a comprehensive efficiency analysis on the EAV dataset. As summarized in Tab. 11, our FMENet achieves the highest accuracy (76.96%) and F1-score (76.47%) while maintaining competitive efficiency across all computational metrics. Specifically, it attains a balanced profile with moderate model size (0.502 MB), computational complexity (1.95 GLOPS), and inference speed (48.15 sample/sec). This optimal balance between state-of-the-art performance and practical efficiency demonstrates that FMENet's superiority stems from its effective architectural design rather than excessive parameterization, making it suitable for real-world deployment scenarios.

We also compared our model with standard EEG classification models, such as EEGNet, in order to quantify the specific increases in the number of parameters, floating-point operations (FLOPs), and end-to-end training time for FMENet+FELB. The results are shown in the table 12. The computational cost of our model is within the acceptable range for practical deployment. The introduced FELB module incurs negligible overhead (0.7K parameters and less than 0.001 GLOPS) and the 15.83 MB frozen VAE decoder only requires initialisation once offline. On a single A5000 GPU, the end-to-end inference latency for producing both the emotion label and the five-frame emoji sequence is approximately 83 ms. This processing speed adheres to the real-time constraints established for HCI applications (100–200 ms) (Butler, 1983). Considering the improvement in performance over the baseline EEGNet (77.13% vs. 58.99%), the modest increase in computational requirements is justified, showcasing the real-time viability and practicality of our approach.

*Table 11.* Computational efficiency comparison of different backbones on the EAV dataset. Darker orange or green indicates better performance for each metric (considering ↑/↓). Our method achieves the best recognition performance while maintaining competitive efficiency.

| Metric | Unit | SyncNet | EEGNet | TSception | ATCNet | EEGConformer | LMDA | EEGMiner | TSLANet | **FMENet (Ours)** |
|---|---|---|---|---|---|---|---|---|---|---|
| Accuracy ↑ | % | 40.18 | 58.99 | 64.40 | 67.68 | 63.33 | 65.12 | 73.10 | 37.80 | 76.96 |
| F1-Score ↑ | % | 37.25 | 61.34 | 63.16 | 66.10 | 61.94 | 64.23 | 72.30 | 37.14 | 76.47 |
| Inference Speed ↑ | sample/sec | 292.6 | 212.56 | 157 | 21.4 | 46.62 | 152.62 | 196.22 | 109.48 | 48.15 |
| Model Size ↓ | MB | 0.001 | 0.018 | 0.789 | 0.284 | 1.733 | 0.023 | 0.003 | 3.413 | 0.502 |
| Parameters ↓ | - | 173 | 4677 | 206807 | 74340 | 454245 | 5992 | 796 | 894215 | 131589 |
| FLOPs ↓ | GLOPS | 0.001 | 0.014 | 0.62 | 2.23 | 1.363 | 0.018 | 0.002 | 2.683 | 1.95 |
| Peak GPU Memory ↓ | GB | 9.95 | 10.03 | 10 | 10.14 | 10.6 | 10.07 | 9.93 | 9.97 | 10.18 |
| Computational Efficiency ↑ | GLOPS/sec | 15.19 | 298.25 | 9740.44 | 477.34 | 6353.43 | 274.35 | 46.86 | 29369.72 | 1506.04 |
| GPU Utilization ↑ | % | 28.29 | 36.2 | 53.54 | 36.56 | 36.38 | 61.23 | 32.7 | 31.17 | 43.25 |

*Table 12.* Computational cost comparison. Cls.=emotion classification; Cls.+Gen.=classification+5-frame emoji generation.

| Model | #Params (M) ↓ | FLOPs (GLOPS) ↓ | Train Time | Infer. Speed (samples/s) | Acc. (EAV, %) ↑ |
|---|---|---|---|---|---|
| EEGNet | 0.005 | 0.014 | ∼4h | 212.6 (Cls.) | 58.99 |
| EEGNet+Our FELB | 0.005 (+0.001) | 0.014 (+0.001) | ∼4.5h | 208.3/52.1 (Cls.+Gen.) | 56.35 |
| EEGConformer | 0.454 | 1.363 | ∼5h | 46.6 (Cls.) | 63.33 |
| EEGConformer+Our FELB | 0.454 (+0.001) | 1.363 (+0.001) | ∼5.5h | 45.8/11.5 (Cls.+Gen.) | 62.75 |
| FMENet (Ours) | 0.132 | 1.95 | ∼5h | 48.2 (Cls.) | 76.96 |
| FMENet+FELB (Our full model) | 0.132 (+0.001) | 1.95 (+0.001) | ∼6h | 47.9/12.0 (Cls.+Gen.) | **77.13** |

## D. Additional Zero-shot Analysis on SEED

In § 4.2 of main paper, we describe how a ResNet-18 classifier is trained on our emoji corpus to provide emotion predictions for generated emojis, which are then compared with frame-level pseudo labels on EAV/MMER. Here, we extend this protocol to the zero-shot SEED setting in order to quantitatively assess how well the emojis generated from SEED EEG remain aligned with SEED's original emotion annotations. Due to issues with channel mismatch, EVA has 30 channels, MMER has 18, and SEED has 62, we adopted a strict 'common channel subset' strategy (Lai-Tan et al., 2025) in our generalisation experiments, to ensure compatibility between the EVA/MMER and SEED datasets. Taking the generalization validation from the EAV dataset to the SEED dataset as an example, we identified 28 electrodes that are physically shared between the two datasets based on the International 10–20 System (Lee et al., 2024; Zheng & Lu, 2015). The model was then retrained using these 28 channels (Fp1, Fp2, F7, F3, Fz, F4, F8, FC5, FC1, FC2, FC6, T7, C3, Cz, C4, T8, CP5, CP1, CP2, CP6, P7, P3, Pz, P4, P8, O1, Oz and O2) from the EAV dataset.

Concretely, we first apply the EAV-trained FMENet to SEED EEG to obtain emoji sequences, and then feed each emoji frame into the ResNet-18 classifier (trained on EAV emojis only). This yields an *emoji-based* emotion prediction for each SEED trial, without any use of SEED facial data or labels during training. We then compare these emoji-based predictions with the ground-truth EEG labels of SEED under the standard 3-way protocol (Positive / Neutral / Negative) and report: **(1) Emoji Accuracy** (%): 3-way classification accuracy of the ResNet-18 predictions on the generated emojis, evaluated against SEED EEG labels. **(2) Label Consistency** (%): the proportion of trials for which the majority vote of emoji-based predictions matches the corresponding SEED label. Per-class and average results are summarized in Tab. 13.

*Table 13.* Zero-shot performance on SEED (Zheng & Lu, 2015) when applying the EAV-trained FMENet and the emoji ResNet-18 classifier without any fine-tuning on SEED. "Emoji Acc." denotes 3-way emotion classification accuracy of the emoji-based predictions w.r.t. SEED EEG labels. "Label Cons." denotes the proportion of trials whose emoji-based prediction agrees with the SEED label.

| Metric | Positive | Neutral | Negative | Avg. |
|---|---|---|---|---|
| Emoji Acc. (%) | 72.31 | 58.37 | 49.58 | 60.09 |
| Label Cons. (%) | 74.15 | 60.23 | 51.07 | 61.82 |

Overall, the emoji-based predictions achieve an average accuracy of about $60\%$, clearly above the $33\%$ chance level, with better alignment on Positive than on Negative trials. Although SEED does not provide human-annotated facial expressions, this indirect comparison between emoji-derived labels and EEG emotion labels suggests that the EEG-to-emoji mapping learned from EAV transfers reasonably well to an unseen EEG-only dataset, and that the generated emojis remain broadly consistent with the underlying affective annotations.

## E. More Case Visualizations

We provide extensive EEG-to-emoji qualitative results on both EAV and MMER datasets in Fig. 13 & Fig. 14. These supplementary visualizations encompass a wider range of emotional states and subject variations, consistently showing high-quality emoji generation with semantically meaningful emotional labels.

*Table 14.* Per-class Precision, Recall, and F1 of the ResNet-18 classifier on the EAV test set (126k frames).

| Class | Support | Precision | Recall | F1 |
|---|---|---|---|---|
| Neutral | 25,658 | 0.82 | 0.82 | 0.82 |
| Happy | 82,542 | 0.93 | 0.89 | 0.91 |
| Sad | 3,992 | 0.55 | 0.74 | 0.63 |
| Angry+Disgust | 6,893 | 0.67 | 0.71 | 0.70 |
| Fear+Surprise | 6,915 | 0.65 | 0.79 | 0.71 |
| Macro Avg. | 126k | 0.72 | 0.79 | 0.75 |
| Weighted Avg. | 126k | 0.86 | 0.86 | 0.86 |
| Accuracy: | | | | 0.855 |

*Table 15.* Confusion matrix of the ResNet-18 classifier (rows: ground truth, columns: predictions).

| GT \ Pred | Neutral | Happy | Sad | Ang+Dis | Fear+Sur |
|---|---|---|---|---|---|
| Neutral | 21010 | 3771 | 296 | 281 | 300 |
| Happy | 3802 | 73462 | 1525 | 1777 | 1976 |
| Sad | 198 | 510 | 2934 | 158 | 192 |
| Ang+Dis | 342 | 871 | 276 | 4941 | 463 |
| Fear+Sur | 296 | 765 | 277 | 138 | 5439 |

**Emotion Label Prediction for Quantifying Semantic Correctness.** To objectively assess the semantic fidelity of the reconstructed emojis, we predict their emotion labels using a dedicated classification model. We fine-tune a ResNet-18 (He et al., 2016) pre-trained for general facial expression recognition on our emoji datasets.

To mitigate the issues of inherent class imbalance and semantic overlap in frame-level facial expressions (for example, the

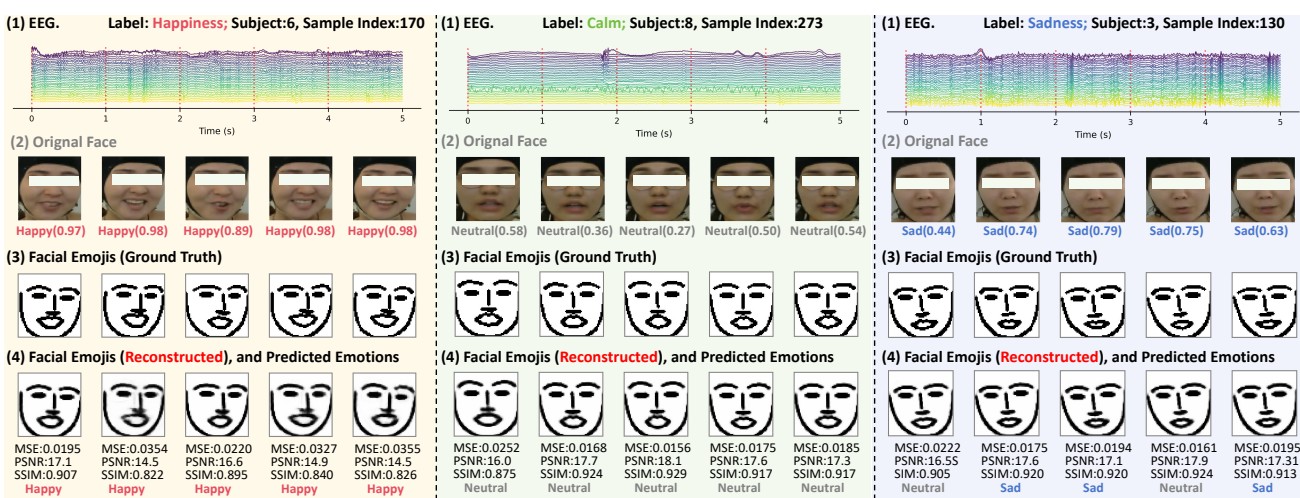

*Figure 13.* Extended visualization of representative examples from the **EAV** dataset.

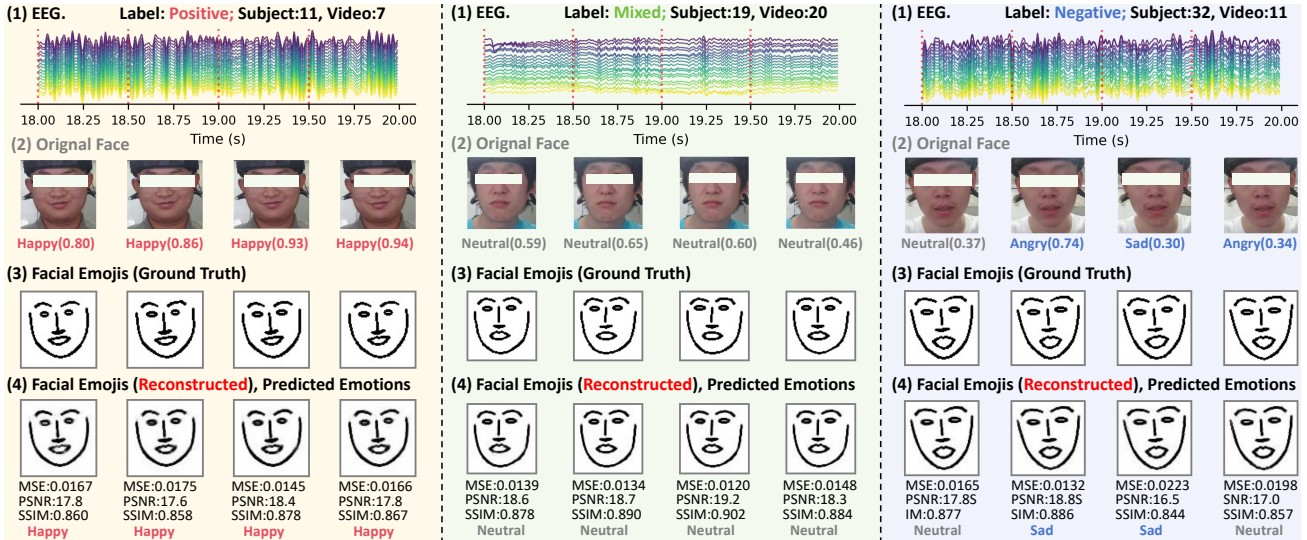

*Figure 14.* Extended visualization of representative examples from the **MMER** dataset.

high similarity in arousal between fear and surprise), we merged semantically similar categories into five meta-classes: neutral, happy, sad, angry+disgust, and fear+surprise. The model was validated using a large-scale test set comprising 126,000 frames from the EAV dataset (our largest benchmark), thereby ensuring robust performance estimates.

**Performance of the ResNet-18 Emotion Classifier.** To ensure that the emotion labels assigned to reconstructed emojis are reliable and free from systematic bias, we provide a detailed performance report of the ResNet-18 secondary classifier. As shown in Tab. 14 and Tab. 15, the classifier achieves an overall accuracy of 85.5% and a weighted F1-score of 0.86. Notably, despite the smaller support for the "Sad" class (Macro F1 = 0.75), the precision/recall balance across merged meta-classes (e.g., Angry+Disgust, Fear+Surprise) indicates no structural bias toward the dominant "Happy" category. Confusion primarily occurs between semantically related expressions (e.g., Neutral vs. Happy), reflecting perceptual ambiguity rather than systematic classifier bias. These results support that the classifier serves as an objective evaluator for emoji semantic fidelity.

The emotion labels displayed beneath each reconstructed emoji in our figures (*e.g.*, row 4 in Figs. 13&14) are the predictions from this model, which serve as the reference for evaluating the semantic alignment between the reconstructions and the underlying EEG emotions.

**Interpretive Value and Temporal Dynamics.** Regarding the conceptual gap between video-level labels and frame-level

emoji generation, the generated emojis are not based on fixed, model-imposed prototypes. As the VAE is trained to reconstruct facial landmarks, it models the geometric variations of expressions present in the dataset. As illustrated in the dynamic sequences (*e.g.*, Fig. 6 right and Fig. 14 most right), a video assigned a single video-level label (*e.g.*, "Negative") can exhibit frame-level transitions (*e.g.*, from neutral to anger, or alternating between sad and anger). This frame-level variation reflects the temporal dynamics of the input rather than a contradiction with the global label. Compared to static video-level predictions, this frame-level output provides additional granularity regarding how facial expressions change over time. Accordingly, the generated emojis are intended as stylized visualizations of facial geometric dynamics, rather than direct proxies for ground-truth psychological states.

**Observations from Extended Cases.** The additional examples reinforce our main findings: (1) Reconstruction quality remains consistently high across diverse emotions (MSE $\approx 0.02$, SSIM $\approx 0.9$); (2) Semantic alignment between reconstructed emojis and EEG emotional labels is maintained even for challenging mixed-emotion scenarios; (3) The structural fidelity of reconstructed emojis ensures that the emotion classifier produces confident and accurate predictions, validating the effectiveness of the EEG's visually interpretable approach.

# F. Impact & Ethical Considerations

This work proposes a framework that visualizes affective information in EEG signals through identity-anonymized facial emojis. By mapping high-dimensional neural activity into a compact geometric space of stylized expressions, the method aims to improve interpretability and privacy compared to direct use of raw facial videos. At the same time, EEG-based emotion technologies raise non-trivial societal, epistemic, and ethical questions. In this appendix, we discuss potential benefits, intrinsic limitations, and normative boundaries of our approach.

## F.1. Positive Impact and Application Prospects

**Clinical and Assistive Applications.** For individuals with difficulties in emotion recognition or expression (*e.g.*, autism spectrum disorder, some forms of aphasia), EEG-driven emoji visualization could serve as a real-time, objective feedback channel. Such a channel may support therapeutic interventions and self-awareness training by externalizing internal affect-related physiology in a simple, readable form. Importantly, our representation is identity-anonymized by design, which aligns with privacy-preserving requirements in many clinical workflows.

**Neurofeedback and Rehabilitation.** In mental health and affect regulation settings, the proposed system can be used as a biofeedback interface. Compared with traditional numeric or curve-based feedback, EEG-to-emoji visualization provides an intuitive depiction of how physiological signals associated with affective states evolve over time. This may make it easier for users to understand, monitor, and gradually modulate their own emotional responses, especially when integrated into structured neurofeedback protocols.

**Enhanced Human–Computer Interaction.** Embedding EEG–emoji proxy modeling into interactive systems may enable more emotionally aware AI agents. For example, a learning assistant could detect signs of frustration or disengagement and adjust its teaching strategy; a companion robot could infer user engagement or affective shifts and adapt its dialogue policy accordingly. Because our model operates on EEG and outputs abstract emojis rather than photorealistic faces, it offers a pathway to increase empathetic responsiveness while limiting direct exposure of biometric identity.

**Neuroscience and Affective Computing Research.** From a research standpoint, the framework provides a computational tool for testing hypotheses about *neural–facial consistency*, i.e., the relationship between internal affective states and facial dynamics across individuals and contexts. By jointly analyzing EEG-driven emoji trajectories, observable behavior, and subjective reports, researchers can probe cross-modal correspondences in affective processing and refine mechanistic models of emotion-related neural dynamics.

## F.2. Inherent Limitations and Risks

Emotions are complex, subjective, and contextually and culturally shaped constructs. Our method deliberately adopts simplified, binary facial emojis as the target representation. This design emphasizes interpretability, privacy, and computational efficiency, but necessarily introduces abstraction and loss of nuance.

**Granularity Trade-offs.** Binary emojis capture only prototypical geometric patterns of facial expressions. They are not designed to represent micro-expressions, subtle intensity variations, or complex blended emotions (*e.g.*, bittersweet, nervous

excitement). Consequently, the generated emojis should be understood as schematic summaries of expression-related motor patterns, not as exhaustive renderings of the underlying affective experience.

**Limited "Truth" Status.** The emojis produced by our framework are *model-based hypotheses* inferred from EEG patterns, not direct readouts of subjective experience or ground-truth affect. They reflect how the model interprets neural signals given its training data and inductive biases. In any downstream use, these visualizations should be treated as auxiliary cues or candidate explanations, rather than objective or definitive evidence about an individual's inner state.

### F.3. Ethical Boundaries and Prohibited Uses

Given the sensitivity of EEG and affective information, the design, training, and deployment of EEG–emotion technologies must adhere to strict ethical principles and governance frameworks.

**Recommended and Legitimate Use Cases.** We consider the following scenarios as broadly aligned with responsible use, provided that robust informed consent and data protection safeguards are in place:

- Clinical and rehabilitation settings, where EEG–emoji visualization is used as a supportive tool for assessment or intervention under professional oversight.

- Voluntary personal health tracking or affect self-management applications, with clear and revocable consent.

- Assistive communication devices supporting people with expression or communication impairments.

- Basic research on affective neuroscience and affective computing, subject to institutional ethics review and appropriate anonymization.

**Strictly Prohibited or Highly Problematic Uses.** We explicitly caution against, and do not endorse, the following applications:

- **Coercive or manipulative contexts:** use in interrogations, security screenings, or any high power-imbalance setting where individuals may feel pressured or forced to undergo EEG-based emotion monitoring, especially when outputs are used to exploit vulnerabilities or exert undue influence.

- **Surveillance and privacy violations:** continuous or covert emotion monitoring in public or private spaces without explicit, informed, and freely given consent, even if only abstract emojis are displayed. Such uses risk infringing on mental privacy and autonomy.

- **Over-reliance in high-stakes decisions:** treating system outputs as the sole or decisive criterion in legal judgments, clinical diagnoses, hiring or firing decisions, or other high-impact determinations. In these domains, EEG-based visualizations should, at most, complement but never replace comprehensive human-led assessments.

We encourage future researchers and practitioners to engage proactively with ethics boards, legal experts, and affected communities when adapting or extending this line of work, and to establish clear usage boundaries, transparency measures, and accountability mechanisms.

## G. Future Work and Responsible Development

Advancing EEG–emoji proxy modeling from proof-of-concept to trustworthy real-world technology requires coordinated progress across methodology, evaluation, and governance. Here we outline several directions for future work with an emphasis on responsible development.

### G.1. User-Centered Evaluation

Most of our current evaluation relies on benchmark datasets and model-centric metrics. Future work should incorporate systematic user studies, including:

- Collaborations with clinicians and rehabilitation specialists to assess usability, interpretability, and clinical utility of emoji visualizations in real workflows.

- Co-design with patients and lay users to identify visualization formats that are intuitive, non-stigmatizing, and minimally prone to misinterpretation or emotional burden.

- Cross-cultural and cross-linguistic studies to examine how different communities interpret emoji-based expressions, and to avoid universalizing a culture-specific expression standard.

## G.2. Bias Auditing and Fairness

Emotion recognition and facial expression analysis are known to be sensitive to demographic factors (*e.g.*, gender, age, ethnicity, neurodiversity). Our datasets (EAV, MMER and SEED) lack race and cultural annotations, and no publicly available EEG-face dataset currently provides such labels. We acknowledge this as an open question. Regarding age distribution, subjects are primarily young adults (18-35 years), meaning age coverage is limited. From another perspective, different demographic groups may exhibit distinct facial landmark distributions, which could be implicitly encoded in emojis—potentially helping group differentiation, though this remains speculative without proper data. Future research should:

- Evaluate our method across more diverse populations as suitable datasets become available.

- Systematically audit performance across subgroups, including both classification accuracy and emoji reconstruction quality.

- Analyze structural biases in training data, such as imbalanced distributions of emotion labels or expression styles, that may lead to systematic misclassification (*e.g.*, misreading neutral expressions as negative in certain groups).

- Explore training strategies with explicit fairness constraints or debiasing mechanisms, aiming to maintain comparable semantic reliability of emoji proxies across populations.

## G.3. Richer Yet Controllable Visual Representations

Within a "privacy- and interpretability-first" design philosophy, it may be beneficial to explore more expressive but still controllable proxy spaces:

- Extending from binary emojis to continuous representations over facial action units or intensity scales, to better capture graded affective changes while preserving anonymity.

- Incorporating temporal regularities (*e.g.*, trajectories of facial action, micro-dynamics) to more faithfully describe the process of emotional unfolding without reconstructing realistic faces.

- Providing adjustable abstraction levels (*e.g.*, toggling between coarse and fine-grained emoji representations) so that different application domains can choose an appropriate privacy–fidelity trade-off.

## G.4. Norms, Guidelines, and Governance

No single study can fully anticipate the societal ramifications of affective AI. We therefore advocate for continued community and regulatory efforts to develop:

- Domain-specific guidelines for EEG-based affective technologies, covering data collection, cross-modal alignment, model training, result presentation, and data sharing.

- Clear minimum standards for informed consent in emotion-related EEG research and applications, including honest communication about capabilities, limitations, and potential risks.

- Procedures for independent auditing and impact assessment, including bias detection, misuse prevention, and safety evaluation, as part of pre-deployment review.

## G.5. Concluding Perspective

We view this work as one step toward more transparent and human-aligned neural interfaces, rather than a definitive solution for "reading emotions from the brain." The ultimate value of EEG–emoji proxy modeling will depend on how it is designed, interpreted, and governed in practice. Only under a cautious and humble attitude—and with human well-being as the central objective—can such technologies serve as tools for empowerment and understanding, rather than instruments of surveillance or control.

