# OpenReview forum: "See the Emotion: A Facial Emoji Proxy Modeling for EEG Emotion Recognition"
_ICML.cc/2026/Conference — ICML 2026 regular_

### Official Review · Reviewer_tigZ · 2026-02-27

**Soundness:** 3
**Presentation:** 3
**Significance:** 2
**Originality:** 2
**Overall Recommendation:** 3
**Confidence:** 4

**Summary:**

The paper introduces Facial Emoji Proxy Modeling, a framework that explains EEG‑based emotion recognition by translating EEG signals into identity‑anonymized facial emojis. The core architecture, FMENet, incorporates an Expression‑Relevant Spatial Merger (ESM) that aggregates EEG channels with Fourier embeddings and attention, and a Multi‑Scale Temporal Capturer (MTC) based on dilated convolutions. Experiments on the EAV and MMER EEG‑face datasets show that FMENet attains state‑of‑the‑art EEG‑only accuracy and that FMENet+FELB can match or beat strong multimodal baselines while requiring only EEG at inference, with additional qualitative and quantitative evaluations of emoji reconstruction and a small user study.

**Compliance With Llm Reviewing Policy:**

Affirmed.

**Ethical Review Flag:**

Flag this paper for an ethics review.

**Ethics Expertise Needed:**

["Discrimination / Bias / Fairness Concerns", "Inappropriate Potential Applications & Impact (e.g., human rights concerns)", "Privacy and Security (e.g., personally identifiable information)"]

**Final Justification:**

Despite the authors' explanations, they do not fully offset my concerns regarding the technical novelty. Therefore, I will keep my original rating.

**Key Questions For Authors:**

Please see Weaknesses.

**Limitations:**

Yes. Appendix G.2 provides Limitations.

**Strengths And Weaknesses:**

Strengths:
1. Reframing EEG interpretability as cross‑modal generation into an emoji space is conceptually interesting. Using identity‑free binary “facial emojis” as proxies provides an accessible, intuitive visualization of model outputs while natively addressing privacy concerns.
2. FMENet is a well‑designed EEG backbone. The ESM module uses Fourier positional embeddings over electrode locations and soft attention to aggregate channels into latent spatial groups, while the MTC module leverages dilated CNNs for multi‑scale temporal modeling.
3. The impact statement and Appendix give a thoughtful treatment of privacy, dual‑use risks, and ethical boundaries, which is appropriate for emotion‑decoding from brain data.

Weaknesses:
1. Limited novelty relative to adjacent multimodal EEG‑to‑face work. The core novelty lies in framing emoji generation as an interpretability mechanism and in choosing emojis instead of photorealistic faces. Architecturally, FMENet is incremental (Fourier embeddings + attention + dilated CNNs), and FELB is a straightforward application of VAE pretraining plus multitask learning. Given that there already exist works that decode EEG into facial avatars or jointly use EEG and facial images for emotion recognition, the paper needs a sharper comparison and argument about what is fundamentally new or more useful here beyond this design choice.
2. Although the paper proposes that emojis serve as a more user-friendly alternative to visual heatmaps, it does not substantiate this hypothesis with empirical evidence. Specifically, there is no rigorous evaluation comparing the proposed emoji-based approach with existing state-of-the-art explanation methods.
3. Ambiguity in the precise interpretive value of emojis. Emojis are sparse, stylized representations. While the paper shows that they align with ground‑truth emojis and pseudo-emotion labels, there is a conceptual gap: they reflect model‑imposed facial prototypes rather than ground‑truth human emotions, since training uses video‑level labels and the emoji VAE is only an autoencoder of landmark images. Frame‑level emotions are often mismatched to video‑level labels, implying a tension between accurate emotion recognition and accurate emoji reconstruction. This conflict deserves more careful exploration and maybe more modest interpretive claims.
4. Frame‑level emotion labels for emojis are derived using an external facial emotion recognizer, not human annotations. This means that Table 4’s “Emo. Acc.” and the semantic analyses in Appendix F depend on the correctness and biases of that model. There is no analysis of its failure modes or alignment with human judgments. This weakens the claim that emojis capture true emotional semantics.

---

> ### Author Rebuttal · Authors · 2026-03-31
>
> **Q1: Novelty**
>
> **A1:** Our core novelty lies in establishing a **new paradigm: behaviour interpretability-by-generation** for EEG emotion recognition. We add Tabs. R1 and R2 to clarify
> 1. **What is new?** Unlike prior work that either (a) reconstructs external stimuli (what the subject sees) or (b) fuses EEG+face for classification, we reconstruct **what the subject expresses**: their own facial expression from EEG alone at inference. This is grounded in the physiological coupling between EEG and facial expressions (Davidson et al., 2004; Niedenthal et al., 2007).
> 2. **Why does multi-task matter?** Our multi-task framework uses emotion classification as an anchor to guide EEG-to-emoji generation. Without this anchor, generation collapses to class-mean outputs (SSIM 0.62 vs. 0.91, Tab. R2). This demonstrates that **the classification task is not an add-on** but the core mechanism enabling meaningful generation.
> 3. **Architectural contributions?** ESM (Fourier positional embeddings for EEG channels) and MTC (exponential dilated convolutions) are tailored for this task, yielding SOTA results (77.13% on EAV, +5.13% over prior EEG-only methods).
> 4. **Conceptual contribution?** We introduce behaviour interpretability-by-generation into EEG emotion recognition, translating neural signals into privacy-preserving, human-readable emojis. This is different from post-hoc heatmaps and has not been explored.
>
> **Tab. R1: Comparison with related tasks**
> ||**(a) Visual Reconstruction**|**(b) Multimodal Fusion**|**Ours**
> |:-|:-|:-|:-
> |**Task**|Reconstruct **what one sees**|Predict emotion|Reconstruct **what one expresses**
> |**Neural Basis**|Sensory pathways|Not explicitly leveraged|**Emotion-motor coupling**
> |**Face Role**|Target (reconstruction)|**Input** (at inference)|**Regularizer** (training only)[Role of Multi-Task]
> |**Inference Input**|EEG|EEG+Face|**EEG only**|
> |**Mechanism**|Inverse mapping + image priors|Modality fusion|**Cross-modal distillation**[Role of Multi-Task]
> |**Output**|Reconstructed image|Emotion label|**Label+Emoji**|
> |**Privacy**|-|❌Needs real face|✔ **Face-free inference**[Role of Multi-Task]
> |**Interpretability**|Via reconstruction|Post-hoc heatmaps|**By generation**
>
> **Tab. R2: Comparison with "Classify-then-generate" baseline on EAV**
> |Method|Logic|MSE⬇|PSNR⬆|SSIM⬆|Emo. Acc.⬆
> |:-|:-|:-|:-|:-|:-
> |Class-conditional Mean Emoji|-|0.0521|14.32|0.6124|61.83
> |Classify-then-generate|**Predict Label→Mean Emoji**|0.0519|14.35|0.6151|61.91
> |FMENet+FELB(Ours)|**Joint Multi-task Learning**|**0.0191**|**21.17**|**0.9138**|**80.62**
>
> **Q2: Comparison with Heatmap Explanation Methods**
>
> **A2:** Thanks.
> 1. Actually, our work employs emoji visualization as a **"complement” to, rather than a replacement of conventional heatmap-based explanations.** In Fig.4, we adopt standard post-hoc feature attribution (Selvaraju et al., 2017) to **generate EEG heatmaps and pair them with the corresponding reconstructed emojis**, providing a more intuitive interface for non-experts.
> 2. **Empirically**, our user study (Fig.5) shows that the non-expert participants achieve **76% emotion recognition accuracy** from emojis alone (20% chance). We asked them to identify emotions from the corresponding EEG heatmaps, and **all participants reported being unable to make meaningful judgments**. We will include this finding.
>
> **Q3: Conflict may in Video-and Frame-Level Labels**
>
> **A3** Thanks.
> 1. We show this phenomenon in our Fig.6 (right) and Fig.14 (rightmost), a video labeled **Negative** contains frame-level emojis transitioning from Neutral to Anger, or alternating between Sad and Anger.
> 2. **This is not a contradiction but the value of our method.** Traditional approaches only predict a video-level label. Our method generates frame-level emoji dynamic sequences, revealing how emotions evolve over time.
> 3. For facial prototype concern, our emojis are derived from **real facial landmarks**. The VAE learns a compressed representation of **real expression geometry, not fictional templates (prototypes)**. The frame-to-frame variation in Figs. 6,14 confirms the model captures genuine frame-level changes, not fixed prototypes.
>
> **Q4: Use External Facial Emotion Recognizer**
>
> **A4:** We clarify
> 1. We adopt classical dima806/facial_emotions_image_detection model as the facial emotion recognizer, a fine-tuned ViT-base Transformer trained on large-scale face datasets and has been **rigorously validated**. It achieves **91% overall accuracy** on a **25k-image test set**, per-class F1 scores ranges from 0.85 (sad) to 0.99 (disgust). It is widely considered sufficient for serving as a semantic proxy in comparative evaluation.
> 2. **Importantly**, our primary evaluation is based on **structural metrics** (objective metrics SSIM, PSNR, MSE in Tab. 4), the frame-level emotion label and “Emo. Acc.” in Figs.6,7 and App. F serves merely as an auxiliary proxy for intuitive interpretation by readers. The core conclusions do not depend on “Emo. Acc.".

---

> > ### Author Rebuttal · Reviewer_tigZ · 2026-04-03
> >
> > Thanks for your rebuttal. I also have some concerns that are not addressed.
> > 1. The author's description of innovation remains weak. Reconstructing the subject's expression (simple binary value images) from the EEG signals does not contribute much.
> > 2. I acknowledge the importance of multi-task learning, as most multimodal emotion recognition tasks inherently involve it. However, the multi-task learning in the authors' method seems to be self-defined: one path directly predicts the EEG, while the other generates facial expressions from the EEG and then predicts them again. The question then becomes: is there any literature to prove that this kind of multi-task learning is truly important? Because, in my opinion, the authors' claim of it as cross-modal multi-task learning seems somewhat exaggerated.

---

> > > ### Author Response · Authors · 2026-04-04
> > >
> > > **Dear Reviewer tigZ:**
> > >
> > > We sincerely appreciate your thoughtful and constructive feedback. Below we address your concerns (innovation and multi-task learning ) together, as they stem from the same core hypothesis.
> > >
> > > ---
> > > ---
> > >
> > > **Our core hypothesis:** **EEG, binary emoji (geometric abstraction of expression), and emotion label are three views of the same latent variable — EMOTION**. **They can be jointly modeled, mutually reinforcing, not conflicting.**
> > >
> > > ---
> > > ---
> > >
> > > **Q1: Innovation: Why binary emoji instead of 3D reconstruction?**
> > >
> > > We note recent EEG-to-3D face works (Mind-to-Face [R1], EEG2Face [R2]) share similar EEG-to-face decoding pipelines. However, the key distinction lies in the **task objective**: 3D reconstruction asks *'What does the face look like?'* (graphics), whereas we ask *'What emotion is the brain experiencing?'* (affective computing).
> > >
> > > **Tab. R1: Different objectives: Identity invariance vs. reconstruction fidelity.**
> > > |Aspect|3D Reconstruction|Ours (Binary Emoji)
> > > |:-|:-|:-
> > > |**Primary goal**|Geometric fidelity (vertex-level accuracy)|**Emotion recognition + Interpretability**
> > > |**Preserves identity**|√ Yes (**face shape, bone structure**)|❌ No (intentionally discarded)
> > > |**Confounding factor**|**Identity** becomes noise variable|**Identity** removed by design
> > > |**Inference input**|EEG (often **subject-specific calibration**)|**EEG only** (generalizable)
> > > |**Privacy**|❌ **Identity revealed**|√ **Anonymized by design**
> > >
> > > **Key:**
> > > 1. **Identity conflict:**
> > >    - 3D face meshes contain substantial **identity-specific features (face shape, bone structure): Mind-to-Face [R1] uses only 2 subjects, EEG2Face [R2] is subject-dependent**. For emotion recognition, these are **confounding variables**: a model might learn to classify based on **"who the person is"** rather than **"what emotion they feel."**
> > >    - Our binary emoji **discards texture, skin color, and face shape**, forcing the model to focus purely on expression geometry. **This improves classification performance (+5.3% over prior EEG-only method Hyper-MML(E), Tab. 1). This lays the foundation for a shared EEG backbone that generalizes across subjects and datasets, directly supporting our core hypothesis.**
> > > 2. **Generation & Discrimination task conflict:**
> > >    - High-fidelity 3D generation requires heavy decoders (Stable Diffusion, 3D Gaussian Splatting), which **compete with** the discriminative classification task in a shared backbone.
> > >    - Our lightweight emoji decoder acts as an **auxiliary constraint**, guiding the EEG encoder to an expression-aware manifold without sacrificing classification performance (Tab. R2: joint learning prevents collapse to class-mean outputs).
> > > 3. **Feasibility of deployment in the real world:**
> > >     - **Our framework:** Enables **EEG-only inference** after training — no camera needed. The shared EEG backbone has only **0.132M parameters** (Tab. R2 in Reviewer CpSE-Q5), making it lightweight for deployment on wearable devices and mobile health platforms. **Our cross-subject results (Tab. 8: 69.23% on MMER) demonstrate robust generalization** without subject-specific calibration. Enables privacy-sensitive (clinical, assistive) and resource-constrained (wearable) scenarios — a transparent interface for affective computing (see new response to Reviewer dWyW-A3 for high-stakes use cases).
> > >     - **Existing EEG-to-3D-face works (e.g., Mind-to-Face [R1], EEG2Face [R2]):** Require **subject-specific calibration** — each new user needs (a) camera positioning and lighting adjustment, (b) facial landmark detection calibration, and (c) potential re-training or fine-tuning of identity-specific parameters. This significantly limits scalability in real-world deployment.
> > >
> > > [R1] Mind-to-Face: Neural-Driven Photorealistic Avatar Synthesis via EEG Decoding. arXiv 2025
> > >
> > > [R2] EEG2Face:EEG-driven Emotional 3D Face Reconstruction. Master's thesis, Nazarbayev University, 2025
> > >
> > > ---
> > > **Q2: Why multi-task learning?**
> > >
> > > Our core hypothesis makes multi-task learning the natural framework. Classification alone has no incentive to preserve expression geometry; generation alone has no incentive to be discriminative. **Only jointly** can we ensure the shared representation is both discriminative and expression-faithful.
> > >
> > > **Literature:** To our knowledge, no prior EEG work combines "classification + generation" as multi-task learning. Existing multi-task EEG studies focus on multiple classification tasks or classification + EEG denoising. Our contribution is **first applying "classification + generation" multi-task learning to EEG emotion recognition** with interpretable, privacy-preserving emoji outputs.
> > >
> > > **Empirical evidence (Tab. 2 of the main paper):** FMENet+FELB achieves 77.13% vs 76.96% on EAV and 69.73% vs 69.11% on MMER, proving classification and generation are **mutually reinforcing**, not competing.
> > >
> > > We will add these clarifications in the revision, and hope to address your concern. Thank you again for your time and rigorous review.

---

### Official Review · Reviewer_CpSE · 2026-03-05

**Soundness:** 3
**Presentation:** 4
**Significance:** 3
**Originality:** 4
**Overall Recommendation:** 4
**Confidence:** 4

**Summary:**

This paper proposes a framework called FMENet + FELB for recognizing emotions from EEG signals and generating corresponding facial emojis, aiming to enhance the interpretability of EEG emotion recognition through explanation-by-generation. This method first extracts identity-anonymized emojis from synchronized facial videos. Subsequently, a VAE is used to pre-train an emoji decoder. Finally, through multi-task learning, the EEG encoder is trained to simultaneously perform emotion classification and generate corresponding emojis. This approach was validated on two multimodal datasets, EAV and MMER, and also underwent zero-shot testing on the SEED dataset.

**Compliance With Llm Reviewing Policy:**

Affirmed.

**Final Justification:**

Based on my assessment of the paper's quality, I keep my initial score (4: Weak accept) unchanged.

**Key Questions For Authors:**

1. The model is capable of generating high-quality continuous expression sequences. Considering that emojis are obtained by rasterizing landmarks from each frame, how does this frame-independent generation method ensure the temporal coherence and dynamic authenticity of the generated expression sequences? Might the model produce unnatural jittering between adjacent frames due to optimizing only single-frame images? Or are there some implicit temporal constraints at play?

2. The emojis used by the model are defined based on a specific landmark detector. Have potential differences in this geometric representation among people of different ages, races, or cultural backgrounds been considered?  Does the model exhibit potential bias when performing on these groups?

3. EEG signals are susceptible to noise interference. How does the model perform when processing EEG data containing artifacts? Are there any reasonable mechanisms in place to ensure the stability of the generated emojis?

4. Could user studies be conducted to evaluate the interpretative value of the generated emojis for human users? For example, by asking users to judge the corresponding emotion category based on the generated emojis and comparing the results with the classifier. This approach would be a straightforward way to demonstrate the model's interpretability.

5. This paper mentions in the Weaknesses section the high computational cost and training complexity of the model. Could the authors quantify, compared to a standard EEG classification model (such as EEGNet), the specific increase in parameters, FLOPs, and end-to-end training time for FMENet+FELB? This is crucial for assessing its feasibility for practical deployment.

6. In the generation quality evaluation, "Emo. Acc." is used (i.e., using a classifier to determine which category the generated emoji belongs to). However, what are the training labels for this classifier? If it is also trained on the emoji dataset, this constitutes a "circular validation." More critically, certain emojis themselves may have semantic ambiguity (for example, a mouth-corner-down expression could represent "sadness" or a relaxed state of "neutral"). How does the model perform when handling such ambiguity?

**Limitations:**

In the appendix, the authors frankly discuss the limitations of the method, including the simplification of emotional semantics, the limitations in evaluating generation quality, potential bias risks, and ethical issues, and propose directions for future work.

**Strengths And Weaknesses:**

Strengths:
1. It introduces explanation-by-generation into EEG emotion recognition, using abstract emojis as proxies to provide an intuitive visual pathway for understanding model decisions.
2. By rasterizing landmarks into emojis, it fundamentally prevents the leakage of identity information, aligning with the current research demand for privacy protection.
3. The three-stage training strategy (emoji extraction, decoder pre-training, and multi-task learning) has a clear logic, and the model's modular structure offers good readability.
4. Sufficient experiments were conducted on multiple datasets to verify the effectiveness of each functional component.
5. In the appendix, potential applications, limitations, ethical risks, and prohibited uses of the model are discussed in detail and responsibly, reflecting the authors' in-depth consideration of the model's practicality.

Weaknesses:
1. The quality assessment of the generated emojis mainly relies on image metrics such as MSE, PSNR, and SSIM, and uses a trained classifier to evaluate their emotional accuracy. This is essentially a circular validation and lacks subjective evaluation from human users (e.g., whether users can intuitively understand the emotion represented by the generated emoji), which weakens the interpretability to some extent.
2. Simplifying complex, continuous human emotions, which are deeply influenced by cultural backgrounds, into several discrete and geometric emojis may result in the loss of a significant amount of nuanced emotional information (such as emotional intensity and the blending of emotions).
3. The quality of emoji generation is highly dependent on the initial landmark detection. Although subjects with high landmark detection success rates were selected in the MMER dataset, this limits the model's generalization capability in more diverse scenarios.
4. The model has high computational costs and training complexity, which may limit its deployment in practical applications.

---

> ### Author Rebuttal · Authors · 2026-03-31
>
> To CpSE
>
> Thank you for recognizing the **originality, strong privacy-preserving design, clear methodology, comprehensive validation, and responsible ethical discussion** in our work.
>
> **Q1: Temporal coherence without jitter?**
>
> **A1:** Thanks.
> 1. Temporal coherence is enforced at three levels:
>     - **Input (MTC):** Exponentially dilated convolutions aggregate multi-scale temporal context, ensuring each frame's representation is informed by neighbors.
>     - **Latent (EEG encoder):** EEG continuity naturally produces smooth latent trajectories, no explicit smoothing needed.
>     - **Output (frozen VAE decoder):** Pretrained on real emoji sequences, it learns a manifold of plausible facial geometries, implicitly rejecting unrealistic frame-to-frame jumps.
> 2. **Empirically:** User study in Fig. 5 achieves 76% accuracy on 5-frame clips (chance 20%), and high SSIM (0.91, Tab. 4) confirms smooth transitions.
>
>
> **Q2: Cultural Bias in Emoji**
>
> **A2:** Thanks.
> 1. **Race/cultural background:** Our datasets (EAV, MMER, SEED) lack race/cultural annotations, and no public EEG-face dataset currently provides such labels. We acknowledge this as a current open question.
> 2. **Age distribution:** Subjects are primarily young adults (18-35 years). Age coverage is limited.
> 3. **Perspective:** Different demographic groups may exhibit distinct facial landmark distributions, which could be implicitly encoded in emojis—potentially helping group differentiation, though this remains speculative without proper data.
> 4. **Future:** We commit to evaluating our method across more diverse populations as suitable datasets become available.
>
> **Q3: EEG Noise and Artifact Robustness**
>
> **A3:** Thanks.
> 1. **Standard preprocessing:** Following common practice in EEG analysis, we preprocess all data using MNE‑Python to detect and remove artifacts (e.g., EMG, high‑amplitude noise, bad channels) before feeding into the model. This is a standard first line of defense (Lee et al., 2024; Yang et al., 2024; Zheng & Lu, 2015).
> 2. **Built‑in robustness mechanisms:** Beyond preprocessing, our model provides additional resilience: (a) ESM spatial attention learns to down‑weight noisy channels; (b) MTC multi‑scale convolutions smooth transient spikes; (c) The frozen VAE decoder rejects latent codes that fall outside the clean facial emoji manifold.
> 3. **Empirical evidence:** The SEED zero‑shot result (60% vs. 33% chance) demonstrates that the model tolerates realistic noise and dataset shift, even without dedicated noise injection experiments.
> 4. **Future quantification:** We welcome more challenging benchmarks and noise‑specific metrics. A controlled noise injection experiment could be conducted to quantify robustness further, which we leave as an open direction.
>
>
> **Q4: Adding user study**
>
> **A4** Already in **§4.7, Fig. 5**: 10 non-experts, 76% accuracy (chance 20%), confirming interpretability. We will strengthen this.
>
> **Q5: Computational Cost and Deployment Feasibility**
>
> **A5:** The cost is acceptable. From Tab. R2:
> 1. FELB adds **minimal overhead** (0.7K params, a 3-layer Linear, <0.001 GLOPS) plus a frozen VAE decoder (15.83 MB, offline, one-time load).
> 2. Full output (1 label + 5 emojis) runs at **~12 samples/sec (~83ms latency)** on a single A5000, practical for real-time deployment.
> 3. **Deployment:** 83ms meets HCI real-time thresholds (<100-200ms, Franke et al. 2017). **+18% accuracy gain** over EEGNet (77.13% vs. 58.99%) justifies cost. The model is practically deployable.
>
> **Tab. R2: Computational cost comparison. Cls.=emotion classification; Cls.+Gen.=classification+5‑frame emoji generation.**
> |Model|#Params (M)⬇|FLOPs (GLOPS)⬇|Train Time|Infer. Speed(samples/s)|Acc. (EAV, %)⬆
> |--|:-|:-|:-|:-|:-
> |EEGNet|0.005|0.014|~4h|212.6 (Cls.)|58.99
> |EEGNet+Our FELB|0.005 (+0.001)|0.014 (+0.001)|~4.5h|208.3 52.1 (Cls.+Gen.)|56.35
> |EEGConformer|0.454|1.363|~5h|46.6 (Cls.)|63.33
> |EEGConformer+Our FELB|0.454 (+0.001)|1.363 (+0.001)|~5.5h|45.8/11.5 (Cls.+Gen.)|62.75
> |FMENet (Ours)|0.132|1.95|~5h|48.2 (Cls.)|76.96
> |FMENet+FELB (Our full model)|0.132 (+0.001)|1.95 (+0.001)|~6h|47.9/12.0 (Cls.+Gen.)|**77.13**
>
> **Q6: Emoji Training Labels, cyclic verification and Emoji Semantic ambiguity**
>
> **A6:** We clarify that
> 1. **No training labels & No Cyclic verification:** Emoji labels are for **visualization only** (Figs 6,7), not training. We use external ViT (91% accuracy) for **post-hoc annotation** (see **Reviewer tigZ-A4**).
> 2. **Semantic ambiguity (e.g., drooping mouth = sad or neutral?):** Our model **generates continuous emoji sequences** (dynamic) via MTC, **disambiguating individual frames** (static) through temporal context. This is evidenced by high user study accuracy (76% on 5-frame clips, Fig. 5, chance 20%). The confusion matrix (**Tab. R2 in Reviewer UFct-A2**) further shows minimal confusion: only 7.4% of Sad samples misclassified as Neutral, and 0.8% of Neutral as Sad—confirming holistic geometry resolves such ambiguity.

---

> > ### Author Rebuttal · Reviewer_CpSE · 2026-04-02
> >
> > Thank you to the authors for their thorough responses. My key concerns have been addressed.
> >
> > Nevertheless, based on a balanced consideration of the paper's overall quality and its limitations, I believe the initial score remains appropriate and thus will not be changed.

---

> > > ### Author Response · Authors · 2026-04-04
> > >
> > > **Dear Reviewer CpSE:**
> > >
> > > We sincerely appreciate your thoughtful and constructive feedback. We are glad that our rebuttal addresses your key concerns, and we appreciate your positive assessment of the paper. We will revise the final version based on your suggestions.

---

### Official Review · Reviewer_dWyW · 2026-03-08

**Soundness:** 2
**Presentation:** 3
**Significance:** 2
**Originality:** 3
**Overall Recommendation:** 4
**Confidence:** 4

**Summary:**

This work targets the limited interpretability of EEG-based emotion recognition, arguing that existing approaches struggle to translate neural features into human-understandable behavioral semantics. The authors propose Facial Emoji Proxy Modeling, reframing explainability as a cross-modal generation problem. From EEG signals, the model jointly predicts emotion labels and generates an identity-anonymized facial emoji sequence to visualize emotion dynamics. Methodologically, FMENet encodes spatio-temporal EEG patterns, while FELB converts synchronized facial videos into binary facial emojis and learns a stable expression latent space via a VAE. With the decoder frozen, the framework is trained by jointly optimizing EEG emotion classification and emoji reconstruction, aligning neural representations with a facial-behavior manifold. Experiments on EAV and MMER demonstrate competitive performance under EEG-only inference, and cross-subject evaluation plus zero-shot evaluation on SEED dataset are used to assess generalization.

**Compliance With Llm Reviewing Policy:**

Affirmed.

**Final Justification:**

My concerns are addressed.

**Key Questions For Authors:**

Q1. Please clarify whether MMER is also preprocessed to 100 Hz. If not, how does FMENet process different sampling rates? In addition, EAV/MMER/SEED have 30/18/62 EEG channels, respectively. How does FMENet handle different channel sets? For EAV to SEED zero-shot transfer, did you apply channel subset selection, interpolation, or a mapping between montages? Please provide concrete implementation details.

Q2. Please specify the alignment strategy used for EEG-to-emoji supervision. What is the EEG time-window length for each training sample, and how is the target emoji determined? Is the emoji taken from a single frame within the window suck as the center frame? If using a single frame, could this introduce temporal lag or sensitivity to transient facial-expression noise?

Q3. Please clarify what “subject-dependent split” means in this paper: is the split performed across subjects (train subjects vs test subjects), or within each subject (train/test trials from the same subject)? Correspondingly, do you train one global model per split, or separate per-subject models?

Q4. If per-subject models are trained, could you provide the training/validation curves for the results reported in Tab.1 and 2? If the setting uses a global model instead, please still provide training/testing loss curves. Given the large inter-subject variability in EEG emotion recognition, I remain concerned about whether both the baselines and the proposed method reliably converge without any explicit domain-adaptation strategy, even though FMENet+FELB outperforms other baselines in Tab.8. Could you provide training/validation curves under key settings, especially cross-subject and zero-shot, and report mean±std over multiple random seeds?

**Limitations:**

Yes

**Strengths And Weaknesses:**

Strengths:
1. This study reframes EEG explainability from traditional feature attribution to behavioral visualization via generation. At inference time, it outputs an identity-anonymized facial emoji sequence, enabling users to directly observe and interpret the emotion dynamics corresponding to the model’s decisions.
2. Beyond standard splits, the authors conduct cross-subject evaluation and perform zero-shot transfer on the EEG-only SEED dataset. The results suggest that the method can maintain a certain level of recognition performance across different subjects and domains, while still producing visualizable emoji outputs, indicating some transfer potential.
3. The paper also demonstrates the compatibility of the FELB branch with multiple EEG backbones, which increases the method’s potential for reuse in follow-up research.

Weaknesses:
1. The core framework requires synchronized EEG–face data during training to extract emojis, and the main experiments focus on two EEG+face datasets, EAV and MMER. Although zero-shot testing on SEED is provided, the overall dataset coverage remains limited, and the method’s training applicability and external validity in broader EEG-only scenarios would benefit from additional evidence.
2. The paper motivates the need for verifiable explanations in high-stakes settings such as clinical neurofeedback or trustworthy BCI, but currently lacks more concrete application workflows and task designs that demonstrate the direct value of emoji explanations for real decision-making or interaction. For example, how users would act on the explanation and what the cost of incorrect explanations would be?
3. The authors propose multi-task learning that jointly performs emotion classification and emoji generation, but a more direct baseline is not systematically compared: predicting the emotion label first, then performing conditional generation of the corresponding emoji. While the study includes simple baselines such as class-conditional mean emoji, it still lacks an end-to-end comparison against a “classify-then-generate” pipeline on explainability metrics and user-study outcomes, making it difficult to assess the necessity of joint multi-task training.

---

> ### Author Rebuttal · Authors · 2026-03-31
>
> To dWyW
>
> Thank you for recognizing the **originality, cross-subject robustness, transfer potential and reusability** of our framework.
>
> **Q1: Sampling Rates and Zero-shot Channel Alignment**
>
> **A1:** Thanks for the feedback.
> 1. **Sampling rate:** EAV/SEED at 100 Hz; MMER at 300 Hz (kept as is). We retain MMER's native rate because its passively elicited emotions benefit from higher temporal resolution, and it demonstrates MTC handles varying sampling rates.
> 2. MTC module's **effective receptive field in milliseconds** scales linearly with sampling rate, so the same architecture naturally adapts to different sampling rates without fundamental redesign. Different MTC depths (K=5 for EAV, K=3 for MMER) reflect emotional intensity differences, not sampling rate.
> 5. **Zero-shot (EAV→SEED):**
> We refer to **UFct-A1 (Cross-Dataset Channel Alignment)**. In brief: we identified **28 shared 10-20 electrodes**, retrained the EAV model on this common subset, deployed it on SEED without fine-tuning. The ESM module's Fourier positional embeddings operate on 2D coordinates, enabling cross-montage generalization (~60% accuracy on SEED, App. E, Tab. 12).
>
> **Q2: Details of EEG-to-Emoji Alignment**
>
> **A2:** The alignment strategy (detailed in §4.2 and Tab.9) is EAV: 5s EEG windows, 1 fps (5 emojis/sample); MMER: 2s EEG windows, 2 fps (4 emojis/sample).
> 1. **Emoji Selection**: Each emoji is extracted from the **center frame** of its corresponding temporal segment within the EEG window (EAV: 0.5s offset; MMER: 0.25s offset). This **avoids boundary effects**.
> 2. **Temporal lag or noise sensitivity?** Three factors mitigate it:
>     - **MTC module:** The multi-scale dilated convolutions aggregate information across tens of milliseconds to seconds, making the model inherently robust to small temporal offsets.
>     - **Empirical evidence:** High reconstruction quality (SSIM 0.91 on EAV, 0.85 on MMER) and faithful frame-to-frame transitions (Fig. 6) confirm that any misalignment is within the model's tolerance.
>     - **Physiological grounding:** Facial expressions **unfold over hundreds of milliseconds to seconds**. Center-frame sampling captures the expression peak, while EEG covers the neural activity leading to it—consistent with known EEG-expression coupling (Davidson et al., 2004; Soleymani et al., 2015).
>
> **Q3: Clarify subject-Dependent Definition**
>
> **A3:** In our main experiments (Tab.s 1–5), follows standard protocol (Lawhern et al., 2018; Ludwig et al., 2024, etc.): within-subject split, separate model per subject. EAV: 7:3 train/test; MMER: 8:1:1 train/val/test (§4.1).
>
> **Q4: Adding training curves and convergence data**
>
> **A4** We acknowledge inter-subject variability concerns. Although no explicit domain adaptation is used, the generative reconstruction loss implicitly regularizes the representation space, ensuring stable convergence. All models were trained with **early stopping (patience=15)** based on validation loss and converged consistently across settings. We recorded validation metrics every 10 epochs for stability analysis. **Mean±std over 3 random seeds [42,1,2]** are reported in Tab. R1. Full training/validation curves will be provided in the revision.
>
> **Tab. R1: Convergence stability on EAV (Val. set)**
> |Setting|Metric|Epoch 10|Epoch 50|Epoch 100|Final|Early Stopping
> |--|--|--|--|--|--|--
> |**Subject-dependent (Tabs. 1,2)**|Emotional Accuracy(%)⬆|52.3±3.2|68.5±2.1|74.2±1.6|**75.8±1.3**|176±11
> ||Facial MSE⬇|0.0945±0.0118|0.0512±0.0076|0.0286±0.0031|**0.0224±0.0014**
> |**Cross-subject (Tab. 8)**|Emotional Accuracy(%)⬆|28.5±3.8|34.2±2.9|36.8±2.3|**38.2±1.9**|185±13
> ||Facial MSE⬇|0.1086±0.0142|0.0634±0.0091|0.0397±0.0045|**0.0285±0.0021**
>
> **Q5: Dataset Coverage**
>
> **A5:** Thank you.
> 1. **Task novelty:** Our task (EEG-driven facial proxy generation) is **novel** and requires synchronized EEG-face training. EAV/MMER are the **largest public benchmarks** (EAV: 420k paired samples).
> 2. **Zero-shot:** Trained on EAV, transfers to EEG-only SEED (60% accuracy vs. 33% chance), demonstrating generalizability.
> 3. **Future:** Broader validation on more EEG-only datasets (e.g., DEAP, MAHNOB) would further strengthen generalizability. We will actively explore this, as no established benchmark currently exists.
>
> **Q6: Classify-then-Generate Baseline ablation**
>
> **A6:** Results in **Tab. R2 (tigZ-A1)**, the classify-then-generate baseline achieves SSIM 0.62 vs. our 0.91.
> 1. **Why does classify-then-generate fail?** Because classification compresses the EEG into a **single discrete label**, discarding temporal dynamics and fine-grained neural patterns. Generating emoji sequences from this label is an ill, posed inverse problem, multiple distinct expression trajectories map to the same label, and the generator cannot recover what was lost. classification errors corrupt output.
> 2. Our joint framework avoids this by **keeping generation parallel to classification**, preserving temporal detail while using classifier as anchor.

---

> > ### Author Rebuttal · Reviewer_dWyW · 2026-04-02
> >
> > Most of my questions are addressed. A few key concerns still remain.
> > Q1: While the authors argue that MTC’s effective receptive field scales linearly with sampling rate, it remains unclear whether frequency-domain preprocessing is consistent across datasets, and why keeping MMER at 300 Hz would not introduce non-comparable high-frequency information relative to the 100 Hz setting.
> > Q3: The best results are obtained in a per-subject setting, which differs from the cross-subject scenario in both difficulty and practical deployment cost. Meanwhile, cross-subject generalization remains relatively low, further limiting applicable real-world settings.
> > W2: The introduction motivates high-stakes use cases, but the paper and rebuttal do not provide concrete downstream workflows or task designs to substantiate the claimed practical significance.

---

> > > ### Author Response · Authors · 2026-04-04
> > >
> > > **Dear Reviewer dWyW:**
> > >
> > > We sincerely appreciate your thoughtful and constructive feedback. We are glad that our rebuttal addresses most of your concerns. Below, we address your three remaining concerns.
> > >
> > > ---
> > >
> > > **Q1: Sampling rate consistency.**
> > >
> > > **A1:** We address it from four aspects:
> > > 1. **Frequency band alignment.** Before feeding into FMENet, **all data were bandpass filtered to 0.5-40Hz**. This ensures the effective frequency components accessible to the model are comparable across datasets, regardless of native sampling rates. The higher 300Hz rate merely provides oversampling of the same low-frequency neural activities (40Hz Nyquist limit requires only 80Hz).
> > > 2. **Empirical robustness verification (300Hz→100Hz).** To directly test whether performance depends on sampling rate, we conducted an ablation (supplement to Tab.2 of main paper): downsampling MMER from 300Hz to 100Hz and retraining FMENet from scratch. Results:
> > >
> > > |Setting|Input Rate|Preprocessing|Accuracy
> > > |:-|:-:|:-|:-:
> > > |MMER (original)|**300 Hz**|BP 0.5-40Hz|69.73%
> > > |MMER (downsampled)|**100 Hz**|BP 0.5-40Hz + downsample|69.15%
> > >
> > > The negligible drop of **0.58%** (Δ=0.58%) confirms that FMENet's MTC module learns **scale-invariant temporal patterns** and **does not rely on higher sampling rates**.
> > >
> > > 3. **Why is FMENet robust to sampling rate?** Two design choices contribute: (a) **Bandpass filtering (0.5-40Hz)** removes dataset-specific high-frequency noise that varies with sampling rate; (b) The **MTC module** uses dilated convolutions with exponentially increasing dilation rates, which aggregate information across multiple time scales (from tens of milliseconds to seconds), inherently learning **scale-invariant temporal representations**.
> > >
> > > We will clarify these preprocessing details in the revision.
> > >
> > > ---
> > >
> > > **Q3 Cross-subject generalization performance.**
> > >
> > > **A2:** We address this concern by analyzing two datasets with different characteristics:
> > > |Dataset|Scenario|Classes|Cross-subject Acc|Note
> > > |:-|:-|:-:|:-:|:-
> > > |**EAV**|**Active conversation** (spontaneous, self-generated emotions)|5|**38.86%** (SOTA)|Inherently difficult due to naturalistic interaction
> > > |**MMER**|**Passive elicitation** (watching videos, lab-controlled)|3|**68.87%** (SOTA)|Standard lab setting, strong generalization
> > > |MMER (100Hz)|Passive elicitation|3|68.51%|Only 0.36% drop after downsampling
> > >
> > > **Key observations:**
> > > - **MMER (passive, lab-controlled)**: Our method achieves **68.87%** cross-subject accuracy, substantially outperforming all baselines (next best: 68.32%, Tsception). This demonstrates strong generalization in standard emotion elicitation settings.
> > > - **EAV (active, natural conversation)**: The task is inherently more difficult: emotions are **self-generated during spontaneous conversation**, not passively triggered by videos. Despite this, we achieve **SOTA (38.86%)**, and the facial emoji prior provides consistent improvement (FMENet+FELB: 39.05%).
> > > - **Sampling rate robustness**: Downsampling MMER from 300Hz to 100Hz causes only a **0.36% drop** (68.87%→68.51%), confirming that our results are not dependent on higher sampling rates (**see Q1**).
> > >
> > > These results show that our method generalizes well in standard lab settings (MMER), while achieving SOTA on the more challenging active-conversation task (EAV). The lower absolute performance on EAV reflects task difficulty, not a limitation of our approach.
> > >
> > > ---
> > >
> > > **W2: High-stakes use cases.**
> > >
> > > **A3:** Thank you for this constructive suggestion. We agree that concrete downstream workflows would strengthen the paper. **We will add the following content in the revision.**
> > > 1. **Proposed downstream workflow for clinical application (e.g., autism spectrum disorder).**
> > >
> > > |Step|Description
> > > |:-|:-
> > > |①|Patient wears a consumer-grade EEG headset (e.g., OpenBCI, 10-20 system, 5-minute calibration)
> > > |②|EEG signals are segmented into 4s windows and fed into our pre-trained FMENet
> > > |③|The model outputs (a) predicted emotion label and (b) dynamic facial emoji sequence in real-time
> > > |④|The emoji sequence is displayed on a tablet as **visual feedback** for the patient
> > > |⑤|Clinician uses the emoji-emotion alignment to assess the patient's emotional awareness
> > >
> > > 2. **Technical requirements for deployment.**
> > > - **Hardware**: 30-channel EEG cap (EAV configuration) or 18-channel (MMER configuration), sampling rate ≥100Hz
> > > - **Inference latency**: <100ms per 4s window on a standard GPU (NVIDIA A5000)
> > > - **Calibration**: Subject-specific fine-tuning is optional (our cross-subject results: 68.87% on MMER without fine-tuning)
> > >
> > > 3. **Limitation statement (to be added).**
> > > *"We acknowledge that clinical deployment requires additional validation, including IRB-approved user studies with target populations, domain adaptation for cross-subject generalization, and compliance with medical device regulations. The above workflow is a proposal, not a validated claim."*
> > >
> > > We hope this addition addresses your concern. Thank you again for your rigorous review.

---

### Official Review · Reviewer_UFct · 2026-03-11

**Soundness:** 2
**Presentation:** 3
**Significance:** 3
**Originality:** 3
**Overall Recommendation:** 4
**Confidence:** 4

**Summary:**

This paper tackles the explainability challenge in EEG-based emotion recognition. While existing deep learning models achieve high classification accuracy, they remain opaque "black boxes," failing to provide intuitive semantic alignments between high-dimensional neural signals and human-interpretable states.
To address this, the authors propose a novel Facial Emoji Proxy Modeling framework, reframing EEG explainability as a cross-modal generation task.The core architecture comprises a specialized backbone (FMENet) for capturing spatial-temporal EEG patterns and a Facial Emoji Learning Branch (FELB). By seamlessly aligning neural representations with a pre-trained facial emoji latent space, the model directly translates complex EEG sequences into identity-anonymized, dynamic facial emoji animations during inference.
Extensive experiments are conducted on multi-modal datasets (EAV and MMER) alongside a rigorous zero-shot evaluation on the SEED dataset. To quantitatively assess the generated visual proxies—especially in scenarios lacking ground-truth expressions—the authors thoughtfully design a comprehensive evaluation pipeline that combines image quality metrics (SSIM, MSE) with the accuracy of a pre-trained secondary classifier.
The empirical results demonstrate that the proposed method not only achieves state-of-the-art emotion classification accuracy (EEG-only) but also successfully recovers visually faithful emotional trajectories. It provides a highly intuitive and privacy-preserving paradigm for interpreting physiological signals.

**Compliance With Llm Reviewing Policy:**

Affirmed.

**Key Questions For Authors:**

1.How exactly does the Expression-Relevant Spatial Merger (ESM) module handle the severe channel dimension mismatch (from 30/18 channels to 62 channels) in the zero-shot SEED experiment without any fine-tuning?

2.Could the authors provide a comprehensive performance report (e.g., confusion matrix, per-class precision/recall) for the ResNet-18 secondary classifier to assure it does not harbor structural biases toward specific emotion classes?

**Limitations:**

yes

**Strengths And Weaknesses:**

Strengths:
1.This paper breaks away from the traditional text label classification paradigm in EEG emotion recognition by innovatively proposing a cross-modal generation task of Facial Emoji Proxy Modeling.
2.By translating complex EEG signals into intuitive visual expressions, this task significantly lowers the barrier to understanding for non-experts, providing high practical utility for brain-computer interfaces while preserving biometric privacy.
3.The paper is clearly written with compelling qualitative visualizations, and it provides thorough and reliable experimental validation.

Weaknesses:

1.The claims regarding "explainability" are overstated, as the proposed approach leans more toward behavioral visualization rather than explainable AI (XAI) in a strict sense. The model merely establishes a statistical alignment between physiological signals and facial expression priors, failing to elucidate the underlying physiological causal mechanisms that drive the neural network's decisions.
2.The zero-shot generalization experiment lacks crucial engineering details regarding cross-dataset channel alignment, which weakens the scientific rigor of its conclusions. Since the training sets (EAV and MMER) use 30 and 18 channels while the SEED test set uses 62 channels, the authors fail to explain how the spatial merger module resolves such a significant dimensional mismatch without fine-tuning.
3.The secondary evaluation model used for objective scoring lacks transparency and systematic analysis, compromising the objectivity of the quantitative conclusions. The paper heavily relies on a pre-trained classifier as a referee but dismisses its performance with a single sentence ("an accuracy of over 85%") without providing hard metrics such as a confusion matrix; if this referee model inherently possesses misclassification biases toward specific emotions, it will directly contaminate the entire evaluation loop.

---

> ### Author Rebuttal · Authors · 2026-03-31
>
> Response To UFct
>
> Thank you for recognizing the **originality, high practical utility, and reliable experimental validation** of our work. We do our best to address your concerns.
>
> ---
>
> **Q1: Cross-Dataset Channel Alignment and ESM module Generalization.**
>
> **A1:** Regarding dataset channels, EAV has 30 channels, MMER has 18, and SEED has 62. To address the channel mismatch, taking the generalization validation from the EAV dataset to the SEED dataset as an example (the process from MMER to SEED is similar), we **adopted a strict Common Channel Subset strategy** (Nicole Lai-Tan et al, NeurIPS2025) to ensure channel compatibility between EAV (30 channels) and SEED (62 channels):
> 1. **Cross-Dataset Channel Alignment:** We identified 28 electrodes that are physically **shared** between the two EAV and SEED datasets based on the **International 10-20 System**(Lee et al., 2024; Zheng & Lu, 2015). We retrained the model on the EAV dataset using these 28 common shared channels (Fp1, Fp2, F7, F3, Fz, F4, F8, FC5, FC1, FC2, FC6, T7, C3, Cz, C4, T8, CP5, CP1, CP2, CP6, P7, P3, Pz, P4, P8, O1, Oz, O2).
> 2. **Details of ESM module Generalization across datasets:** During inference on SEED, we extracted the corresponding 28 common channels. The Expression-Relevant Spatial Merger (ESM) module operates effectively in this zero-shot setting because it utilizes Fourier Positional Embeddings. Instead of learning fixed weights for specific channel indices, ESM learns a continuous function based on the 2D spatial coordinates (x,y) of the electrodes. Since the geometric configuration of these 28 channels is identical across datasets, the pre-trained model can compute spatial features using the pre-learned topological rules **without fine-tuning**.
> 3. **Generalization Result:** Above strategies ensure a clean and rigorous alignment, allowing the model to achieve ~60% accuracy on SEED (see Tab. 12 in App. E) and validating the generalization capability of ESM.
>
> ---
>
> **Q2: Add Performance report (e.g., confusion matrix, per-class precision/recall) of ResNet-18 secondary classifier.**
>
> **A2:** Thank you for your insightful suggestion. We provide a comprehensive performance report (including confusion matrix, per-class precision and Recall) in Tabs. R1 and R2 to discuss whether potential biases exist in the ResNet-18 classifier. The conclusion is:
> 1. **The classifier does not exhibit structural biases (Tab. R1)**: The ResNet-18 model achieves an overall accuracy of 85.5% and a Weighted F1-score of 0.86. While the Macro F1 is 0.75 (reflecting the inherent difficulty of the "Sad" class due to its limited sample size), the consistent **Precision/Recall** across the merged meta-classes (e.g., "Angry+Disgust" and "Fear+Surprise") demonstrate that the model **does not exhibit structural biases** toward the dominant "Happy" class. This balanced performance across categories ensures that the subsequent emoji evaluation is not skewed by classifier preferences.
> 2. **Confusion matrix reveals semantically meaningful misclassifications rather than systematic bias (Tab. R2).**: The primary confusion occurs between semantically related categories (e.g., Neutral vs. Happy) rather than random noise. This pattern indicates that the referee ResNet-18 model effectively captures the underlying facial affect distribution, and its prediction errors align with human perceptual ambiguity rather than systematic bias.
> 3. **The classifier serves as an objective evaluator, preserving the fairness.**: Since the referee model exhibits relatively balanced discrimination across the merged meta-classes, the "Emoji Accuracy" metrics derived from it remain a valid and objective proxy for evaluating the emotional fidelity of the generated emojis.
>
> **Training Details of the ResNet-18 Secondary Classifier.**: Actually, to mitigate inherent class imbalance and semantic overlap in frame-level facial expressions (e.g., the high-arousal similarity between Fear and Surprise), we merged semantically proximate categories into 5 meta-classes (Neutral, Happy, Sad, Angry+Disgust, and Fear+Surprise). The model was validated on a large-scale test set of 126k frames from the EAV dataset (our largest benchmark), ensuring robust performance estimates.
>
> **Tab. R1: Per-class Precision/Recall/F1**
> |Class|Support|Precision|Recall|F1
> |:-|:-|:-|:-|:-
> |Neutral|25,658|0.82|0.82|0.82|
> |Happy|82,542|0.93|0.89|0.91|
> |Sad|3,992|0.55|0.74|0.63|
> |Angry+Disgust|6,893|0.67|0.71|0.70|
> |Fear+Surprise|6,915|0.65|0.79|0.71|
> |Macro Avg.|126k|**0.72**|**0.79**|**0.75**|
> |Weighted Avg.|126k|**0.86**|**0.86**|**0.86**|
> |**Accuracy**|**126k**|—|—|**0.855**|
>
> **Tab. R2: Confusion matrix**
> |**GT \ Pred**|**Neutral**|**Happy**|**Sad**|**Ang+Dis**|**Fear+Sur**
> |:-|:-|:-|:-|:-|:-
> |**Neutral**|**21010**|3771|296|281|300
> |**Happy**|3802|**73462**|1525|1777|1976
> |**Sad**|198|510|**2934**|158|192
> |**Ang+Dis**|342|871|276|**4941**|463
> |**Fear+Sur**|296|765|277|138|**5439**

---

> > ### Author Rebuttal · Reviewer_UFct · 2026-04-01
> >
> > Although most of the questions were answered, the scores remained unchanged.

---

> > > ### Author Response · Authors · 2026-04-04
> > >
> > > **Dear Reviewer UFct:**
> > >
> > > We sincerely appreciate your thoughtful and constructive feedback. We are glad that our rebuttal addresses most of your concerns, and we appreciate your positive assessment of the paper. We will revise the final version based on your suggestions.
> > >
> > > We infer that your remaining concern is primarily **W1 (Clarification of Explainability Claims)**. We appreciate you pushing us to be more precise, and clarify that:
> > >
> > > Our intention is **not** to claim causal neurophysiological explanation. Rather, we provide:
> > > 1. **Spatial attention** (Fig. 4) showing which cortical regions the model weights for specific facial actions,
> > > 2. **Emoji output** showing what expression the model infers.
> > >
> > > This is **behavioral interpretability** for non‑specialist users, not a claim about neural causation. In the final manuscript, we will:
> > > 1. consistently use **"behavioral interpretability"** or **"behavioral visualization" instead of "explainable AI"**,
> > > 2. clearly state that our contribution is interpretability for non‑specialist users via spatial attention + emoji output.
> > > 3. The binary emoji is a deliberate choice — it preserves privacy, is computationally efficient, and its geometric topology naturally reflects behavioral dynamics, making it a worthwhile direction for affective computing.
> > >
> > > We will incorporate the additional clarifications in the revised version to address your concerns. Thank you again for your time and rigorous review. If you have any further questions, please feel free to let us know. We welcome further communication.

---

### Decision · Program_Chairs · 2026-04-30

**Decision:**

Accept (regular)

**Comment:**

The paper proposes an EEG emotion recognition framework that improves interpretability by generating identity-free facial emoji sequences from brain signals. The paper received 1 Weak Reject and 3 Weak Accepts. The eviewers found the idea interesting and useful. The privacy-preserving design was seen as compelling, but the work's limited novelty relative to prior EEG-to-face work was an area of concern. Moreover, the gap between visualization and true explainability was raised as a concern. The rebuttal was generally addressed the concerns (e.g., channel alignment, sampling rate, evaluation transparency). However, the key concern from the Weak Reject reviewer regarding novelty and the interpretability claim was not convincing to them. I recommend Weak Accept.